# Phytohormone cytokinin guides microtubule dynamics during cell progression from proliferative to differentiated stage

Juan Carlos Montesinos[1], Anas Abuzeineh[2], Aglaja Kopf[1], Alba Juanes-Garcia[1], Krisztina Ötvös[1,3], Jan Petrášek[4], Michael Sixt[1] & Eva Benková[1,*]

## Abstract

Cell production and differentiation for the acquisition of specific functions are key features of living systems. The dynamic network of cellular microtubules provides the necessary platform to accommodate processes associated with the transition of cells through the individual phases of cytogenesis. Here, we show that the plant hormone cytokinin fine-tunes the activity of the microtubular cytoskeleton during cell differentiation and counteracts microtubular rearrangements driven by the hormone auxin. The endogenous upward gradient of cytokinin activity along the longitudinal growth axis in *Arabidopsis thaliana* roots correlates with robust rearrangements of the microtubule cytoskeleton in epidermal cells progressing from the proliferative to the differentiation stage. Controlled increases in cytokinin activity result in premature reorganization of the microtubule network from transversal to an oblique disposition in cells prior to their differentiation, whereas attenuated hormone perception delays cytoskeleton conversion into a configuration typical for differentiated cells. Intriguingly, cytokinin can interfere with microtubules also in animal cells, such as leukocytes, suggesting that a cytokinin-sensitive control pathway for the microtubular cytoskeleton may be at least partially conserved between plant and animal cells.

**Keywords** cell differentiation; cytokinin; cytoskeleton; microtubules; microtubules dynamics
**Subject Categories** Cell Adhesion, Polarity & Cytoskeleton; Plant Biology
**The EMBO Journal (2020) 39: e104238**

## Introduction

Growth and development of living organisms depend on the constant production of new cells that subsequently differentiate,

thereby acquiring specific shapes and functions. The cytoskeleton provides an elementary framework for cell functions, including cell division, cell motility, cell shape, intracellular organization, and trafficking of organelles (Brandizzi & Wasteneys, 2013; Akhmanova & Steinmetz, 2015). Microtubules (MTs) as a major component of the eukaryotic cytoskeleton play important roles in virtually every aspect of its functions. To fulfill these diverse activities, MTs assemble into distinct arrays that are characterized by high dynamics (Horio & Murata, 2014).

In plants, specialized tissues, called meristems, maintain the proliferative capacity and constantly produce new cells, thereby undergoing a gradual transformation from a proliferative to a fully differentiated stage. Typically, the progress of cells through the individual phases is spatio-temporally tightly controlled, resulting in the formation of discrete domains encompassing proliferation, transition, expansion/elongation, and differentiation in plant organs (Le *et al*, 2004; Hayashi *et al*, 2013). As cells progress from the proliferative to the differentiation stage, the microtubular cytoskeleton goes through substantial rearrangements to accommodate cyto-physiological changes that occur during cytogenesis. In dividing cells, MTs are involved in the formation of the preprophase band (PPB) in the cell equator, which predicts the future orientation of the division plane (de Keijzer *et al*, 2014; Hashimoto, 2015). MTs, as a component of the mitotic spindle, contribute to the chromosome separation and they participate in the formation of the cell plate that will separate two daughter cells (Hamada, 2014; Smertenko *et al*, 2017). In non-dividing cells, MTs are localized in the cell cortex, designated cortical microtubules (CMTs), and form arrays that are laterally anchored to the plasma membrane (Lucas & Shaw, 2008; Oda, 2015). As plant cells expand, differentiate, and acquire specific shapes, CMTs have an important function in the delivery and deposition of new cell wall components and in the cell shape maintenance (Hashimoto, 2015; Elliott & Shaw, 2018). Mutants in the core subunits of MTs or factors regulating the dynamics and arrangements of the microtubular cytoskeleton exhibit severe defects in cell functionality

1   Institute of Science and Technology Austria (IST Austria), Klosterneuburg, Austria
2   Department of Plant Biotechnology and Bioinformatics, Ghent University and Center for Plant Systems Biology, VIB, Gent, Belgium
3   Bioresources Unit, Center for Health & Bioresources, AIT Austrian Institute of Technology GmbH, Tulln, Austria
4   Institute of Experimental Botany, The Czech Academy of Sciences, Praha, Czech Republic
    *Corresponding author. Tel: +43 2243 9000 5301; E-mail: eva.benkova@ist.ac.at

(Bao *et al*, 2001; Bichet *et al*, 2001; Burk *et al*, 2001; Ishida & Hashimoto, 2007; Ishida *et al*, 2007; Samakovli & Komis, 2017; Panteris *et al*, 2018).

Plant hormones and their complex regulatory networks steer all aspects of plant growth and development (Petricka *et al*, 2012), among which auxin and cytokinin are key hormonal regulators of cell division and differentiation. Both hormones are required to maintain the proliferative activity of cells in suspension cultures (Skoog & Miller, 1957). *In planta*, the auxin–cytokinin crosstalk has a crucial morphogenetic function in the post-embryonic initiation and formation of new organs, such as lateral roots, shoots, leaves, or flowers, as well as in the control of the organization and activity of shoot and root apical meristems (Dello Ioio *et al*, 2008; Ruzicka *et al*, 2009; Marhavý *et al*, 2011; Chandler & Werr, 2015; Schaller *et al*, 2015). The interplay of auxin and cytokinin in the regulation of the root patterning is well described. The ratio of the auxin-to-cytokinin activities along the longitudinal root growth axis determines the cellular progression through distinct phases of the cytogenesis, thereby defining the size of the meristematic zone, the timing and dynamics of cell transition to elongation and the differentiation phase (Billou *et al*, 2005; Dello Ioio *et al*, 2008; Ruzicka *et al*, 2009; Takatsuka & Umeda, 2014; Di Mambro *et al*, 2017). However, the contribution of auxin and cytokinin in the regulation of the microtubular cytoskeleton activity associated with the cell cytogenesis is scarcely understood. Recently, rapid auxin-triggered rearrangements of the MT network in the root epidermal and hypocotyl cells have been reported (Nick *et al*, 1992; Takesue & Shibaoka, 1998; Takahashi *et al*, 2003; Le *et al*, 2005; Vineyard *et al*, 2013; Chen *et al*, 2014; True & Shaw, 2020). In addition, auxin has been proposed to coordinate reorganization of the cytoskeleton in the pericycle and endodermis during early phases of the lateral root organogenesis (Vilches Barro *et al*, 2019). Whereas the auxin-mediated reconfiguration of CMTs in hypocotyls seems to be an indirect consequence of the enhanced cell expansion, the direct auxin effect on the CMT cytoskeleton has not been excluded in roots (Adamowski *et al*, 2019). Unlike the auxin interaction, that of cytokinin with CMT cytoskeleton has not been assessed thus far.

Here, we demonstrate that the cytokinin pathway plays a role in the fine-tuning of the CMT arrangements and dynamics during cytogenesis. In *Arabidopsis thaliana* roots, a gradual increase of the cytokinin activity along the root longitudinal axis correlates with altered dynamics of the CMT cytoskeleton in root epidermal cells. Modulation of the cytokinin activity gradient by either cytokinin supply or modulation of cytokinin perception and signaling dramatically affects the dynamics of the MT cytoskeleton and interferes with the auxin-driven rearrangements of CMTs. However, compared to the rapid auxin effects, the cytokinin-mediated reconfiguration of CMTs is slower, suggesting that auxin and cytokinin might target different pathways that regulate the microtubular cytoskeleton activity. This is further supported by the observation that cytokinin affects MTs in animal cells as well, whereas the auxin effect is restricted to the regulation of plant MTs. However, whether cytokinin interferes with MTs through a regulatory pathway that might be partially evolutionarily conserved between animal and plant kingdoms remains to be resolved.

# Results

## Orientation and dynamics of CMTs change along the longitudinal root growth axis

Root growth results from the steady production of new cells at the root apical meristem and from their gradual expansion. As cells exit the meristematic zone, they proceed through the transition zone, where they lose their proliferation capacity before undergoing a rapid expansion and differentiation (Baluška *et al*, 2010; Schaller *et al*, 2015) (Fig 1A). Throughout all the growth phases, CMTs have an essential function as a framework for the coordinated deposition of new cell wall material and cell shape maintenance (Elliott & Shaw, 2018). Although the core functions of the microtubular cytoskeleton across various cell types and phases of the cytogenesis are largely conserved, CMTs are highly dynamic and their activity is fine-tuned to accommodate plasticity of cell growth. CMTs in cells of different growth zones have distinct patterns, indicating that during the transition from the proliferative to the differentiation phase CMTs might undergo robust rearrangements (Sugimoto *et al*, 2000; Le *et al*, 2004; Oda, 2015). To capture the dynamics of the CMT network in the course of cell growth, we used a live track imaging of root epidermal cells using the vertical-stage confocal microscopy (Movie EV1), thus avoiding perturbations caused by gravity-induced changes in roots.

To correlate the CMT activity with distinct phases of cell growth, we defined root zones based on criteria described in previous reports (Verbelen *et al*, 2006; Ivanov & Dubrovsky, 2013; Slovak *et al*, 2016; Pavelescu *et al*, 2018). In the transition zone (TZ), cells start to elongate until they reach a length of ~ 45 μm; the elongation zone (EZ) includes cells longer than 45 μm until they enter the differentiation zone (DZ) characterized by initiation of root hairs (Fig 1A). To minimize variability, all analyses of CMTs were done in the atrichoblast cell file, i.e., epidermal cells that do not form root hairs, unless mentioned differently (Fig EV1A). To visualize CMTs, we employed the microtubule reporters MAP4-GFP (Marc *et al*, 1998); mCherry-TUA5 (Gutierrez *et al*, 2009); and immunocytochemistry with α-tubulin-specific antibodies. The Fiji software (https://fiji.sc/) was applied to quantify orientation by scoring the proportions of CMTs in certain orientations. In our assays, transversal and longitudinal orientations corresponded to an angle of 90° and 0° between CMTs and the longitudinal root growth axis, respectively (Fig 1C).

In epidermal cells of the TZ and the EZ, CMTs were arranged transversally (84.42 ± 1.1° and 84.45 ± 0.9°, respectively) and their orientation remained unchanged for 60 min. In the DZ, the cellular elongation growth ceased and the CMTs changed from a transversal to an oblique orientation, reaching an angle ~ 51.55 ± 1.5° (Fig 1B and C; Movie EV1; Table EV1). Although the MAP4-GFP marker is an excellent tool to visualize CMT cytoskeleton, it has limitations due to the chimeric origin of the microtubular binding part of the protein (Marc *et al*, 1998). Importantly, visualization of CMTs with α-tubulin-specific antibodies corroborated patterns of CMTs in distinct root growth zones as detected with the MAP4-GFP reporter (Fig EV1B).

The CMT network is highly dynamic and individual CMTs alternate between growing and shortening phases, enabling their quick assembly and disassembly (Horio & Murata, 2014). To gain insights

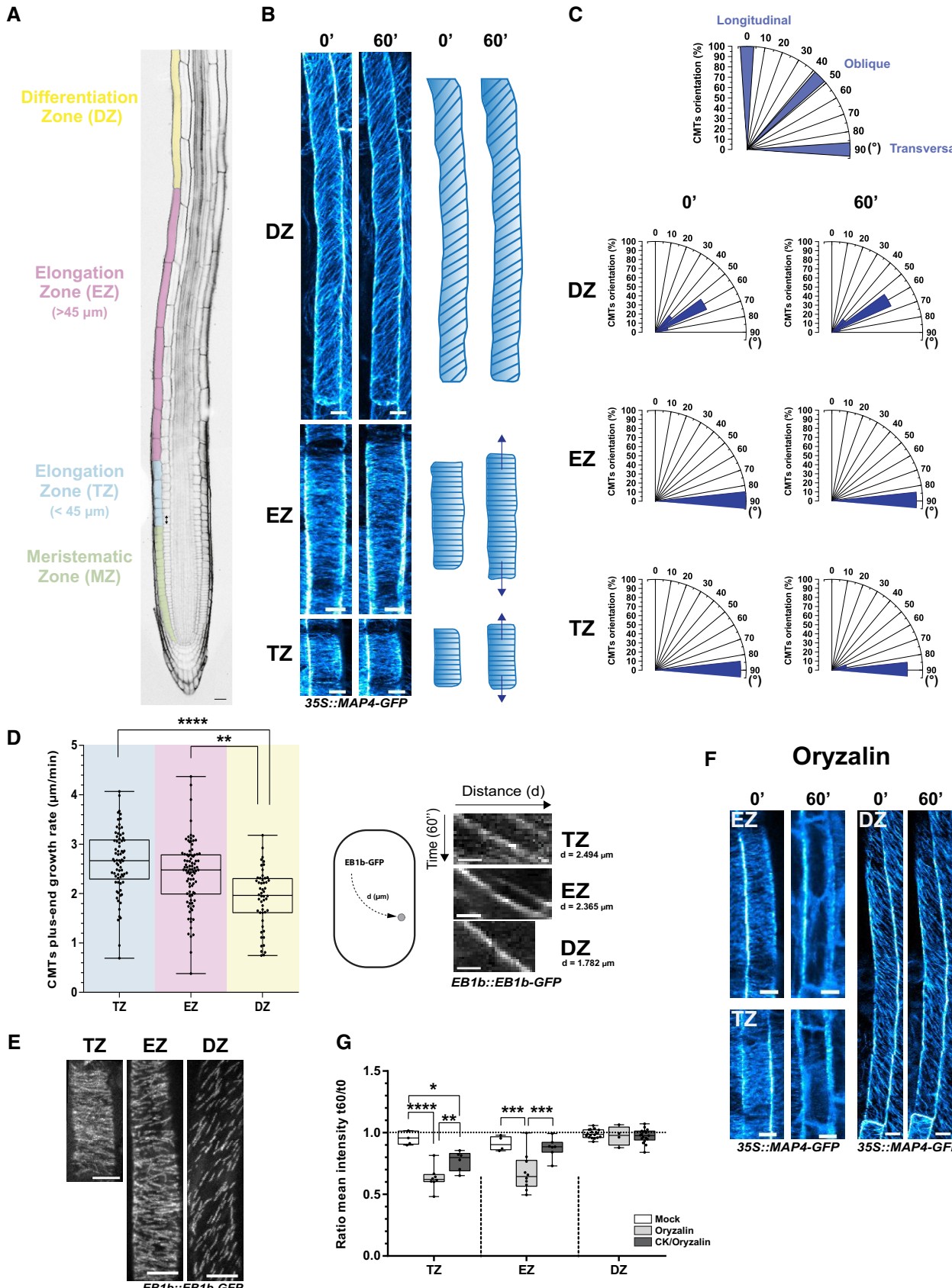

**Figure 1.**

◄

**Figure 1.  Monitoring of cortical microtubules (CMTs) in root epidermal cells.**

A   Root tip of *Arabidopsis* with distinct growth zones marked: meristematic (MZ, green), transition (TZ, blue), elongation (EZ, pink), and differentiation (DZ, yellow). Double arrow indicates the first expanded cortex cell considered as a start point of the TZ, which encompasses epidermal cells smaller than 45 μm. Epidermal cells reaching a length of more than 45 μm prior termination of the elongation are in the EZ. Scale bar 25 μm.

B   CMTs visualized by the MAP4-GFP reporter (left) and scheme of the CMT orientations (right) in epidermal cells of distinct growth zones (TZ, EZ, and DZ). Individual cells monitored at two time points (0 and 60 min). Dark blue arrows mark the cell expansion direction. Scale bar 10 μm.

C   Histograms of the CMT orientation distributions. Orientation was measured as an angle between CMTs and the longitudinal root growth axis, with 0°, 45°, and 90° corresponding to longitudinal, oblique, and transversal orientations, respectively. The proportion of CMTs in a certain orientation is calculated per cell (*n* = 15–20 cells per root growth zone with six–nine roots in four independent replicates).

D   Analysis of the CMT plus-end growth with the EB1b-GFP reporter. CMT plus-end growth rates (μm/min) measured by tracking the EB1b-GFP marker for 20–30 min in epidermal root cells of three growth zones. In the boxplots, the center lines show medians; box limits indicate the 25th and 75th percentiles as determined by the GraphPad software; whiskers span minimum to maximum values; and individual data points are represented by dots. **$P < 0.01$, ****$P < 0.0001$ by Student's *t*-test. The number of CMT plus-end events (EB1b-GFP) tracked per cell was 20–100/min, of which the growth rate average is represented as a single dot; three–five cells per root growth zone, three–five roots analyzed per biological replicate in three independent experiments. On the right, single trajectories of EB1b-GFP signal tracked over 60 s in root epidermal cells at different growth zones. Scale bar 0.5 μm.

E   *Z*-stack maximum image projection of the EB1b-GFP plus-end reporter tracked over 30 s, in epidermal cells at different growth zones. Scale bars 10 μm.

F   CMTs visualized by MAP4-GFP in root epidermal cells at the TZ, the EZ, and the DZ prior (0) and 60 min after treatment with 1 μm oryzalin. Scale bar 10 μm.

G   Quantification of the MAP4-GFP CMT reporter signal in wild-type root epidermal cells at the TZ, the EZ, and the DZ treated with mock (DMSO, white box), oryzalin (1 μM, light gray box), and cytokinin (CK; 10 μM 6-benzylaminopurine, BAP) and oryzalin (1 μM) (dark gray box). For double treatments, roots pretreated for 60 min with CK prior to transfer to medium supplemented with both compounds. Boxplots represent ratio between mean fluorescence intensity (arbitrary units) measured in epidermal cells at 60 and 0 min. The center lines show the medians, and the box limits indicate the 25th and 75th percentiles; whiskers span the minimal to maximal values, and individual data points are represented by dots. Ratio close to 1 (segmented line) corresponds to the unchanged MAP-GFP signal for 60 min (*$P < 0.05$, **$P < 0.01$, ***$P < 0.001$, ****$P < 0.0001$ by Student's *t*-test, *n* = 3–10 cells per root growth zone with five–eight roots per condition in four independent replicates).

into dynamics of CMTs in distinct root growth zones, we used the EB1b-GFP marker to monitor growth rate at plus-end of CMTs (Buschmann *et al*, 2010; Wong & Hashimoto, 2017). By means of spinning disk microscopy, the EB1b-GFP signal was followed for 20–30 min and the CMT growth rates were extracted from time-lapse image sequences. In the epidermal cells at the TZ and the EZ, CMTs plus-end growth rates reached average values of 2.6 ± 0.07 and 2.4 ± 0.07 μm/min, respectively (Fig 1D, Movie EV2). In cells of the DZ, the CMT plus-end growth rate was significantly lower (1.8 ± 0.08 μm/min) than that of less differentiated cells at the TZ and the EZ (Fig 1D, Movie EV3), suggesting that the growth of CMTs at plus-end might cease in cells undergoing differentiation. Visualization of the growth trajectories of CMTs at plus-end by the maximum projection of EB1b-GFP monitored for 30 s supports the transversal orientation of CMTs in cells at the TZ and the EZ, but slanting in the DZ, in line with the MAP4-GFP pattern (Figs 1E and EV1C and D; Movies EV2 and EV3).

To explore whether the dynamics of CMTs adapt to the cellular differentiation status, we tested the CMT sensitivity to oryzalin. This drug sequesters free dimers of tubulin and prevents their addition to the CMT plus-ends, thereby triggering the rapid depolymerization of CMTs (Morejohn & Fosket, 1991; Hugdahl & Morejohn, 1993). It has been shown that CMTs stabilized by taxol are only partially sensitive to oryzalin (Morejohn *et al*, 1987; Hugdahl & Morejohn, 1993). Thus, we hypothesized that differences in the CMT dynamics, as suggested by analyses of microtubule growth rate at plus-ends in cells of distinct root zones, might be manifested by changed sensitivity of CMTs to oryzalin. Oryzalin applied for 60 min led to a quick depletion of CMTs in epidermal cells at the TZ and the EZ, but in cells of the DZ, CMTs were largely insensitive to oryzalin (Fig 1F and G), hinting at changed dynamics of CMTs in cells of the DZ when compared to the TZ and the EZ.

Hence, as cells progress through distinct root growth zones, the CMT network undergoes a pronounced reconfiguration. Transversal positioning and enhanced growth rate at plus-end of CMTs might provide optimal arrangements for the effective deposition of cell wall components in growing cells at the TZ and the EZ when compared to CMTs in differentiated cells, in which the reduced plus-end growth rate and reorientation of CMTs to an oblique orientation correlate with termination of cell expansion.

**Cytokinin and auxin form distinct response gradients along the longitudinal axis of the *Arabidopsis* root**

Coordinated, spatio-temporally controlled transition of root cells through the proliferation and expansion phases until the fully differentiated stage is acquired defines the overall kinetics of the primary root growth (Verbelen *et al*, 2006; Dello Ioio *et al*, 2008; Petricka *et al*, 2012). Auxin and cytokinin are among essential endogenous regulatory molecules, of which the mutually antagonistic activities at the distal root tip have been shown to control the balance between the cell proliferation rate and the transition to elongation and differentiation (Dello Ioio *et al*, 2008; Moubayidin *et al*, 2009; Petersson *et al*, 2009; Antoniadi & Pla, 2015; Di Mambro *et al*, 2017). The distinct patterns and dynamics of CMTs detected in cells of the different root growth zones prompted us to thoroughly monitor the activity of the auxin and cytokinin pathways along the longitudinal root growth axis. To closely examine the balance between the auxin–cytokinin responses in individual cells, we used the novel biosensor *TCSn::ntdT:tNOS-DR5v2:3nGFP* (Smet *et al*, 2019). In agreement with previous reports, we detected mutually complementary, partially overlapping expression patterns the sensitive reporters of auxin (*DR5v2:3nGFP*) and cytokinin (*TCSn::ntdT:tNOS*) in the provasculature, stem cell niche, columella, and lateral root cap (Bishopp *et al*, 2011; Bielach *et al*, 2012; Zürcher *et al*, 2013; Sozzani & Iyer-Pascuzzi, 2014) (Fig 2A). Expression analyses of *DR5v2:3GFP* and *TCSn::ntdT:tNOS* in epidermis along the root growth axis revealed distinct response patterns of these two hormonal pathways (Fig 2A and B). While increase of *DR5v2:3nGFP* along the root axis followed a relatively shallow gradient, the *TCSn::ntdT:tNOS* expression profile exhibited gradual increase across the TZ and the EZ toward the DZ (Fig 2A and B).

Noteworthy, profiles of auxin and cytokinin responses in the epidermal cell files that give rise to root hairs (trichoblasts) differed from those observed in non-root hair cells (atrichoblasts) (Fig EV2A–C). Whereas in atrichoblasts the increase in cytokinin responses could be detected early after the transition into the differentiation phase, in neighboring trichoblasts the auxin responses prevailed and the cytokinin responses increased only after the root hairs were fully formed (Fig EV2A–D). Thus, analyses of the *TCSn::ntdT:tNOS-DR5v2:3nGFP* reporter confirmed the previously reported pattern of the auxin and cytokinin activities at the root apical meristem (Bielach *et al*, 2012) and revealed that the transition from proliferation to differentiation in epidermal cells is accompanied with gradual enhancement of cytokinin activity.

## A cytokinin activity gradient fine-tunes the pattern and dynamics of CMTs along the longitudinal root growth axis

Cytokinin responses increase significantly from the TZ toward the DZ. To test the causality between the cytokinin activity and the arrangements and dynamics of CMTs in cells of distinct growth zones, we analyzed the impact of the increased cytokinin signaling on CMTs in cells at the TZ and the EZ. Intriguingly, within 60 min, cytokinin promoted the reorientation of CMTs from transversal (90°) to oblique ($64.08 \pm 4.6°$ and $56.93 \pm 5.4°$ in the TZ and the EZ, respectively), thus leading to the CMT network configuration typically observed in fully differentiated cells (compare Fig 2C and G with Fig 1B and C; Fig EV1B; Table EV1). Accordingly, visualization of CMT plus-end growth trajectories by the maximum projection of EB1b-GFP monitored for 30 s in cells at the EZ confirmed the cytokinin-driven reorientation of CMTs to an oblique orientation (Fig 2H). Furthermore, measurements based on the real-time tracking of EB1b-GFP signals revealed a significant decrease of the CMT plus-end growth rate in cells of the TZ and the EZ exposed to cytokinin when compared to mock-treated control. Noteworthy, cytokinin reduced the growth rate of CMTs at plus-end in cells of the TZ and the EZ to values comparable with these detected in the epidermal cells of the DZ in mock conditions (Figs 2I and, EV1D and E, Movie EV4).

To further validate cytokinin effects on CMTs, which were observed using reporters derived from microtubules associated proteins, such as MAP4-GFP and EB1-GFP, we also employed mCherry marker fused to α-tubulin 5 isoform (TUA5), one of the building blocks of the CMT cytoskeleton (Gutierrez *et al*, 2009). Cytokinin effects on CMTs visualized by mCherry-TUA5 were evaluated using kymograph in combination with KymoButler tool (Jakobs *et al*, 2019). In accordance with our previous findings, we detected significantly slower growth of CMTs after treatment with cytokinin ($1.70 \pm 0.03$ μm/min) when compared to mock conditions ($2.30 \pm 0.03$ μm/min) (Fig EV1H compared to Fig 2I). Thus, the increased cytokinin activity reconfigures the growth and arrangements of CMTs in cells at the TZ and the EZ to the pattern observed in cells at the DZ.

Altered dynamics of CMTs in cells at the DZ when compared to those at the TZ and the EZ correlated with an increased tolerance to oryzalin-triggered depolymerization (Fig 1F and G). Hence, we tested whether cytokinin might mediate the modulation of the CMT network in cells of the TZ and the EZ that results in the reduced sensitivity to oryzalin, thus mimicking the CMT network configuration in differentiated cells. Unlike mock-pretreated epidermal cells at the TZ and the EZ, in which oryzalin triggered a massive depletion of CMTs (Fig 1F and G), in cytokinin-pretreated cells, the CMTs maintained a largely intact structure (Fig 2D). This observation was in line with our assumption that cytokinin-driven reconfiguration of CMTs reduces their sensitivity to oryzalin (Figs 1G and 2D). To evaluate the long-term impact of cytokinin on the CMT network, 5-day-old seedlings were transferred to medium containing either oryzalin or oryzalin together with cytokinin. In roots exposed to oryzalin for 72 h, a typical swelling of cells occurred at the root tip and particularly in the TZ (Baskin *et al*, 1994) (Fig EV3A). Roots exposed to oryzalin in combination with cytokinin were significantly less affected and, in contrast to roots treated with oryzalin, the CMTs were not completely disassembled (Fig EV3A and B). Importantly, the growth rates of roots treated with oryzalin and oryzalin plus cytokinin did not significantly differ, although both were severely impaired when compared to mock conditions (Fig EV3C).

Finally, we examined whether rearrangement of CMTs to an oblique disposition promoted by cytokinin can be reverted after removal of cytokinin. To this end, CMTs were monitored in roots pretreated for 1 h with either mock or cytokinin and afterward transferred to cytokinin-free (mock) medium. We found that reorientation of CMTs in cells at the EZ triggered by cytokinin is not reversible and microtubules remain in oblique orientation during 5 h of observation. Neither recovery of root growth was observed after removal of cytokinins, hinting at non-transient changes of CMTs driven by cytokinins that might correlate with premature transition of cells to differentiation stage (Fig EV1I and J; Movies EV7 and EV8).

Altogether, our observations reveal that cytokinin fine-tunes the CMT network dynamics, and indicate that its graded distribution along the longitudinal root axis might contribute to the adjustment of the CMT activities as cells progress through distinct phases of cytogenesis.

## The cytokinin receptors contribute to fine-tuning CMT pattern and dynamics

In *Arabidopsis*, the cytokinin signal is perceived by three receptors that belong to the histidine kinase (AHK) family (To & Kieber, 2008) exhibiting distinct, partially overlapping pattern of expression. Analyses implementing transcriptional reporter constructs (Nishimura *et al*, 2004) as well as tissue specific transcriptome profiling pointed at *AHK3* and *AHK4* as a potential receptors mediating cytokinin perception in epidermal cells at the TZ toward the DZ (http://bar.utoronto.ca/efp/cgi-bin/efpWeb.cgi?dataSource = Root).

To test whether the cytokinin-mediated control of the CMT network requires functional receptors, we introgressed the MAP4-GFP reporter into mutants defective in the activity of AHK4/CRE1, AHK3, and AHK2, respectively (Inoue *et al*, 2001; To & Kieber, 2008). In mock-treated roots of *ahk4/cre1-12,* similarly to the wild type, the transversal orientation of CMTs in epidermal cells at the TZ and the EZ prevailed (Fig 3A; Table EV1). However, in contrast to the wild type, in *ahk4/cre1-12* roots cytokinin was unable to trigger the reorientation of the CMTs from the transversal to the oblique position in cells at the TZ and the EZ, and CMTs persisted without significant changes in the transversal orientation after 60-min

treatment (compare Fig 3B and E, and Fig 2C and G; Table EV1). Visualization of CMTs in wild type and *ahk4/cre1-12* with α-tubulin-specific antibodies validated the results obtained by monitoring of the MAP4-GFP reporter (compare Appendix Fig S1A to Fig EV1B).

In addition, the contribution of the AHK4/CRE1-mediated cytokinin signaling in the regulation of the CMT dynamics was inspected by means of the EB1b-GFP reporter. Under control (mock) conditions, no significant differences in the CMT plus-end growth rates between

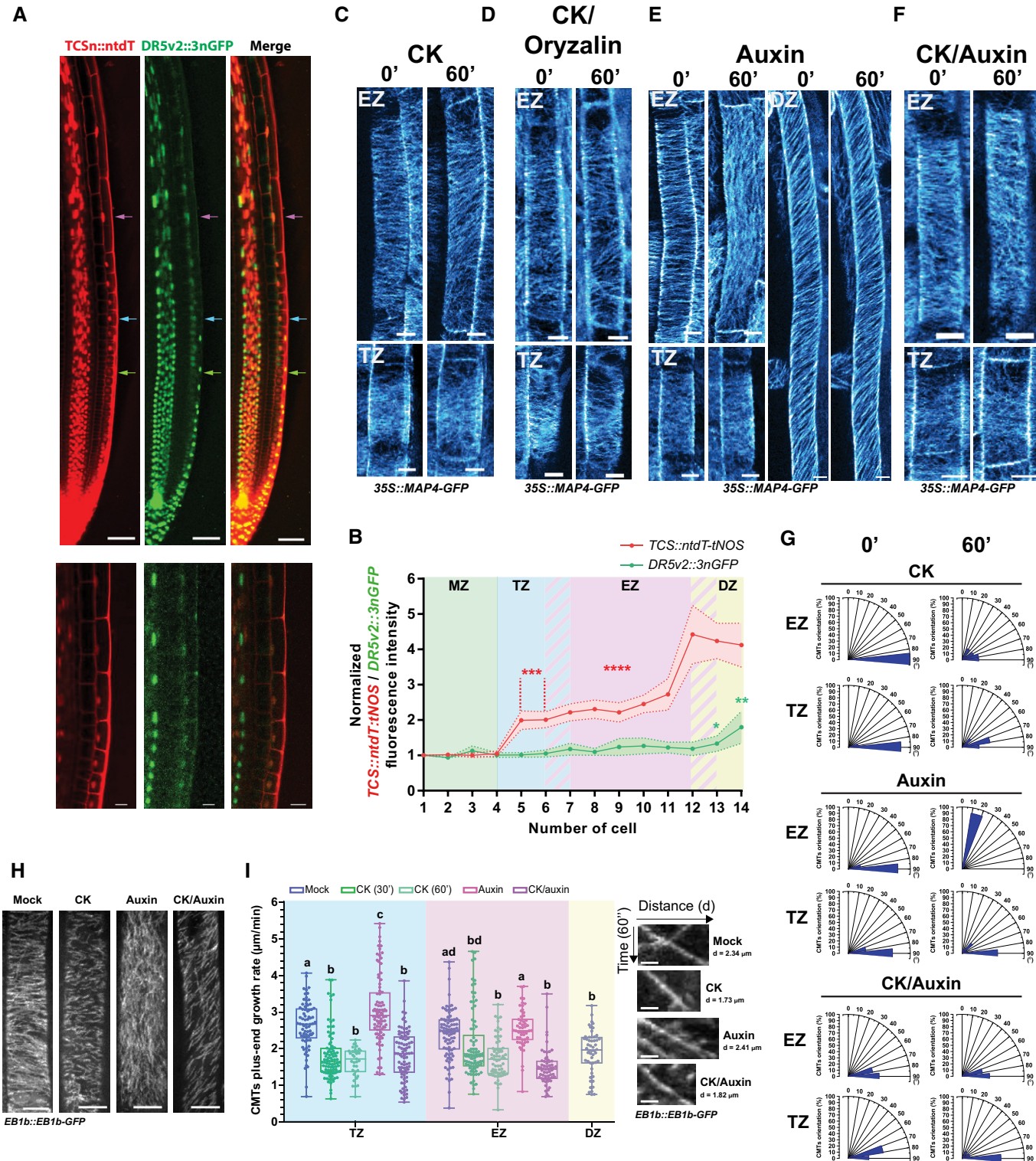

**Figure 2.**

**Figure 2. Cytokinin modulates CMT patterns and dynamics.**

A   *Arabidopsis* root expressing the dual reporter *TCSn::ntdT:tNOS-DR5v2::3nGFP* that is sensitive to cytokinin (CK) and auxin. CK (red), auxin (green), and overlay of both signals detected in nuclei of cells at the root tip (top, scale bar 50 μm). Green, pink, and blue arrowheads point at epidermal cells located in the meristematic zone (MZ), transition zone (TZ), and elongation zone (EZ), respectively. Magnification of the TZ shown (bottom, scale bar 10 μm).

B   Relative fluorescence intensity of *TCSn::ntdT:tNOS* signal (red) and *DR5v2:3nGFP* signal (green) measured in epidermal cells along the longitudinal root growth axis. Cell number 1 corresponds to a meristematic cell placed at position −4 before the beginning of TZ (as marked by green arrow at 2A). Mean ± s.d., *P < 0.05, **P < 0.01, ***P < 0.001, ****P < 0.0001 by Student's *t*-test, *n* = 18 roots.

C–F CMTs visualized with MAP4-GFP in root epidermal cells of the TZ and the EZ at time points 0 and 60 min after CK (C), CK and oryzalin (D), auxin (E), and CK and auxin (F) treatments, and in DZ cells at time points 0 and 60 min after auxin treatment (E). As CK and auxin sources, 10 μM BAP and 0.1 μM NAA were used, respectively, and 1 μM oryzalin. For the double (CK/oryzalin and CK/auxin) treatments, roots were pretreated with CK for 60 min prior to transfer to medium supplemented with both compounds. Scale bar 10 μm.

G   Histograms of CMT orientation distributions in epidermal cells of the TZ and the EZ at time points 0 and 60 min after hormonal treatments as described above (C, E and F). *n* = 15–20 cells per root growth zone with five–eight roots per condition in four independent replicates.

H   Analysis of the CMT plus-end growth with the EB1b-GFP reporter. *Z*-stack maximum image projections of EB1b-GFP tracked over 30 s in epidermal cells without (mock, DMSO) and with CK, auxin, or CK and auxin treatment (60 min). Scale bars 10 μm.

I   CMT plus-end growth rates (μm/min) measured by tracking of EB1b-GFP reporter over 20–30 min in epidermal cells of different growth zones treated for 60 min with mock (DMSO), or media supplemented with CK (30 or 60 min), auxin, and CK plus auxin. In the boxplots, the center lines show the medians; box limits indicate the 25th and 75th percentiles as determined by the GraphPad software; whiskers span minimum to maximum values; and individual data points are represented by dots. The number of CMT plus-end events (EB1b-GFP) tracked per cell was 20–100/min, of which the growth rate average is represented as a single dot. Statistical significance evaluated by two-way ANOVA (*n* = 3–5 cells per root growth zone with three–five roots per replicate in three independent experiments). On the right, single trajectories of EB1b-GFP signal tracked over 60 s in root epidermal cells at different growth zones and treatment conditions. Scale bar 0.5 μm. Hormone concentrations and treatment conditions (H, I) were as described (C, E and F).

the *ahk4/cre1-12* and the wild type could be detected. However, unlike in a wild-type background, cytokinin failed to reduce the CMT plus-end growth rates in the *ahk4/cre1-12* mutant (Fig 3F and G), indicating that the cytokinin-mediated control of the CMT network requires a functional AHK4/CRE1 receptor.

To further investigate the role of AHK4/CRE1 in fine-tuning the CMT network, we tested the sensitivity of CMTs to the oryzalin-triggered depolymerization in *ahk4/cre1-12*. Similarly as in the wild-type background, also in the *ahk4/cre1-12* mutant oryzalin fully disassembled the CMTs in epidermal cells at the TZ and the EZ (Fig 3H and J). However, in cells at the DZ, in which the wild-type CMTs were highly tolerant to oryzalin (Fig 1F and G), lack of the AHK4/CRE1 activity enhanced the sensitivity of CMTs to this drug (Fig 3H and J). Pretreatment with cytokinin, which prevented depolymerization of CMTs by oryzalin in wild-type roots, was unable to attenuate the drug effects in the *ahk4/cre1-12* mutant and CMTs were fully depleted in cells of the TZ and the EZ (compare Fig 3I to Fig 2D, compare Fig 3J to Fig 1G). The typically swollen root tip phenotype occurring after a long-term exposure of the wild-type cells to oryzalin was also visible in the *ahk4/cre1-12* roots. However, in contrast to the wild-type background, co-treatment with cytokinin did not prevent oryzalin-triggered swelling of cells in the TZ and the EZ of the *ahk4/cre1-12* roots (Fig EV3A and B). Growth rates of *ahk4/cre1-12* roots after transfer to media supplemented with either oryzalin or oryzalin plus cytokinin did not significantly differ from those of the wild type. Thus, the more pronounced swelling of cells in the *ahk4/cre1-12* mutant after the oryzalin plus cytokinin treatment than that in the wild type is probably not the consequence of altered growth (Fig EV3C). These results suggest that the AHK4/CRE1 receptor might contribute to the configuration of CMTs along the longitudinal root growth axis and in particular to the adjustment of CMT pattern and dynamics under fluctuating cytokinin levels.

To test whether besides AHK4, its homologues AHK2 and AHK3 might have redundant functions in the regulation of CMTs, the *ahk2-2* and *ahk3-3* mutants were analyzed. Visualization of CMTs using either the MAP4-GFP reporter or α-tubulin antibodies revealed

that under mock conditions, the orientation of CMTs in the *ahk2-2* and *ahk3-3* mutants was comparable to that observed in wild-type epidermal cells at the TZ and the EZ, and that they remained transversal for 60 min (Appendix Figs S1B and C and, S2A and E; Table EV1).

In both the *ahk2-2* and *ahk3-3* mutants, cytokinin promoted the reorientation of CMTs from transversal to oblique in cells of the TZ and the EZ, although the CMT rearrangements in *ahk2-2* were less pronounced than those in the wild-type roots (Appendix Figs S1B and C, and S2B and F; Table EV1). Similarly to the wild-type background, in both the *ahk2-2* and *ahk3-3* mutants oryzalin triggered depolymerization of CMTs in cells at the TZ and the EZ, whereas in cells at the DZ CMTs remained largely unaffected (Appendix Fig S3A, B and D). Pretreatment with cytokinin attenuated the depletion of CMTs (Appendix Fig S3C and D) as well as the swelling of root cells by oryzalin in the *ahk2-2* and *ahk3-3* mutants (Appendix Fig S4A), indicating that the loss of the cytokinin perception through either AHK2 or AHK3 does not dramatically affect the sensitivity of CMTs to cytokinin. Similarly to the wild type, the relative growth rates of *ahk2-2* and *ahk3-3* roots after transfer to media supplemented with either oryzalin or oryzalin plus cytokinin were not significantly different (Appendix Fig S4B).

Next we examined whether lack of cytokinin perception might interfere with pattern of CMT cytoskeleton along the longitudinal root growth axis. Based on correlation observed between cytokinin response and the CMT arrangements along the longitudinal root growth axis, we hypothesized that attenuated cytokinin perception might also affect progression of cells with transversal orientation of CMTs at the TZ and the EZ to differentiated status characterized by oblique disposition of the CMTs. Based on reported expression patterns, we focused on AHK3 and AHK4 as a potentially major receptors acting in these parts of root. We found that while in *ahk4/ cre1-12* mutant, number of root epidermal cells with transversal disposition of the CMTs was not different from wild-type control (10.13 ± 1.11 cells and 10.12 ± 0.95 cells; Fig EV4A), in the *ahk3-3* mutant significantly higher number of cells with transversal CMT orientation (11.92 ± 1.09 epidermal cells) when compared to

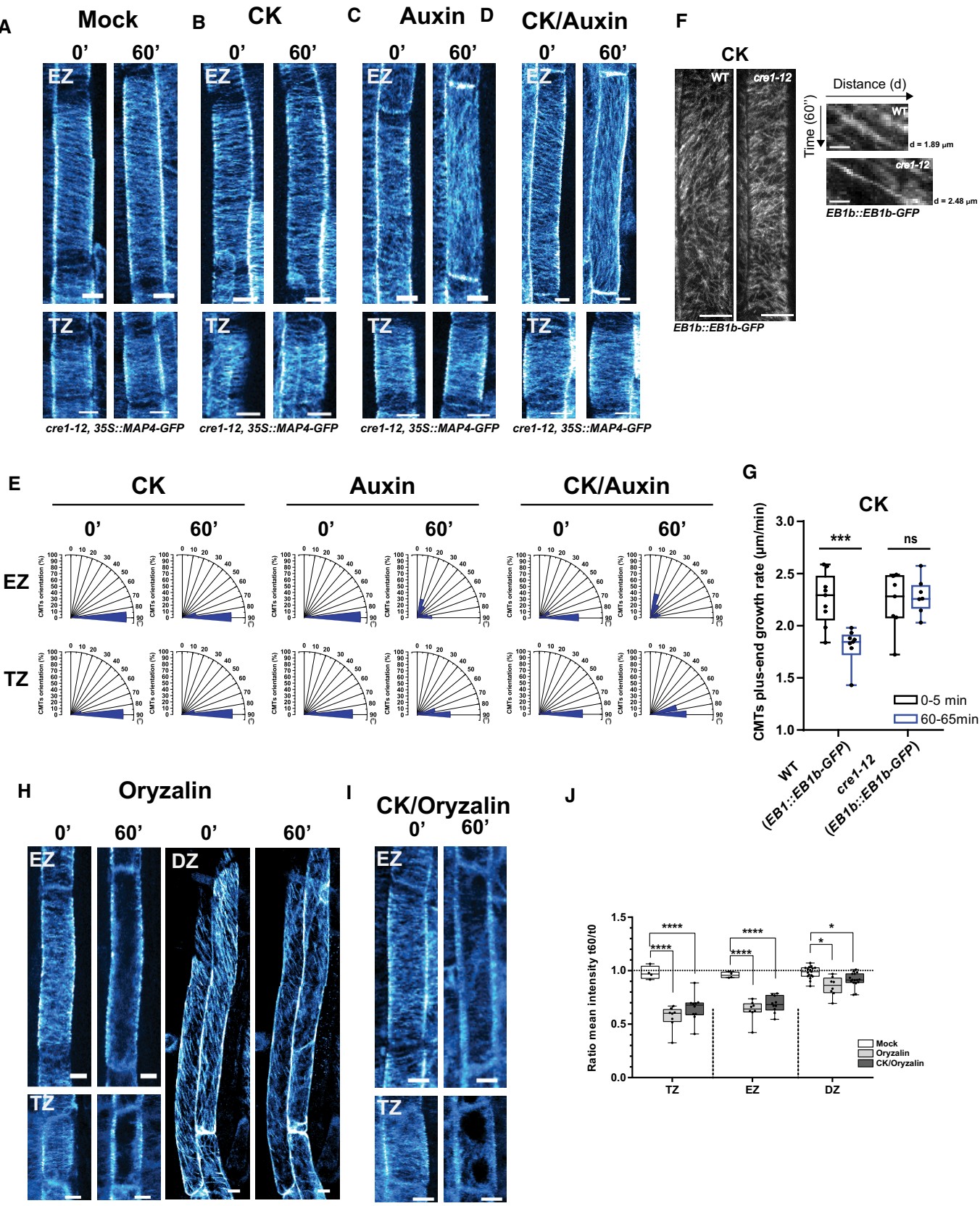

**Figure 3.**

◀ 

**Figure 3. The AHK4/CRE1 receptor mediates the cytokinin effects on CMTs.**

A–D   CMTs visualized with MAP4–GFP in root epidermal cells of the *cre1-12* mutant at the transition zone (TZ) and the elongation zone (EZ) at time points 0 and 60 min after mock (DMSO) (A), cytokinin (CK) (B), auxin (C), and CK plus auxin (D) treatments. As CK and auxin sources, 10 μM BAP and 0.1 μM NAA were used, respectively. For the double CK plus auxin treatment, roots were pretreated for 60 min with CK, whereafter they were transferred to medium supplemented with both compounds. Scale bar 10 μm.

E   Histograms of CMT orientation distributions at time points 0 and 60 min in epidermal cells of *cre1-12* at the TZ and the EZ treated as for (A–D). $n = 14$–29 cells per root growth zone in four–eight roots per condition in three independent replicates.

F, G   Analysis of the CMT plus-end growth with the EB1b-GFP reporter in *cre1-12*. (F) *Z*-stack maximum image projection of EB1b-GFP tracked for 30 s (on the left) and single trajectories of EB1b-GFP signal tracked over 60 s (on the right) in root epidermal cells of the EZ in the wild type and *cre1-12* treated with CK for 60 min. Scale bars 10 and 0.5 μm, respectively. (G) CMT plus-end growth rates (μm/min) quantified from the EB1b-GFP trajectories tracked during 5 min in epidermal cells of the EZ treated with mock (DMSO) or CK. In the boxplots, the center lines show the medians; box limits indicate the $25^{th}$ and $75^{th}$ percentiles as determined by the GraphPad software; whiskers span minimum to maximum values; and individual data points are represented by dots (ns, non-significant, ***$P < 0.001$ by Student's *t*-test, $n = 7$–9 cells with seven roots per condition in three independent experiments). Concentrations and treatment conditions (F, G) were as for (A, B).

H, I   CMTs visualized with MAP4-GFP in root epidermal cells of the *cre1-12* mutant. Cells at the TZ, the EZ, and the DZ were monitored at time points 0 and 60 min after treatment with oryzalin (1 μM) (H) and CK plus oryzalin (I). As CK source, 10 μM BAP was used and 1 μM oryzalin. For double (CK and oryzalin) treatment, roots were pretreated for 60 min with CK and then transferred to medium supplemented with both compounds. Scale bar 10 μm.

J   Quantification of the MAP4-GFP CMT reporter signal in *cre1-12* root epidermal cells at different root zones treated with mock (DMSO, white box), oryzalin (1 μM, light gray box), and CK (10 μM BAP) plus oryzalin (1 μM) (dark gray box). For double treatments, roots pretreated for 60 min with CK prior to transfer to medium supplemented with both compounds. Boxplots represent ratio between mean fluorescence intensity (arbitrary units) measured in epidermal cells at 60 and 0 min. The center lines show the medians, and the box limits indicate the $25^{th}$ and $75^{th}$ percentiles; whiskers span the minimal to maximal values, and individual data points are represented by dots. Ratio close to 1 (segmented line) corresponds to the unchanged MAP-GFP signal for 60 min (*$P < 0.05$, ****$P < 0.0001$ by Student's *t*-test, $n = 3$–10 cells per root growth zone with five–eight roots per condition in four independent replicates).

wild type (Fig EV4A) was detected. Furthermore, when as an indicator of the DZ bulging of root hairs was considered, as expected the CMTs in the atrichoblast cell file in wild-type roots exhibited oblique disposition (41.69 ± 15.07 degrees) and similarly, no significant changes in the CMT orientation in *ahk4/cre1-12* when compared to wild type were detected (41.09 ± 13.53 degrees; Fig EV4B and C). Interestingly, in the *ahk3-3* the CMTs with average orientation 51.96 ± 18.75 degrees was significantly different from wild type suggesting more transversal orientation (Fig EV4B and C).

Altogether, these results hint at the role of cytokinin receptors in regulation of the CMT network. We hypothesize that AHK3 perception might contribute to fine-tuning of the CMT cytoskeleton activity along the longitudinal root growth axis, while AHK4 receptor might play a more prominent role under conditions, which might lead to fluctuations in endogenous levels of cytokinins (e.g., under different type of stresses) (Hare *et al*, 1997; Argueso *et al*, 2009; O'Brien & Benková, 2013).

### The cytokinin signaling is involved in fine-tuning of the CMT network

In *Arabidopsis*, the type-B response regulators (ARRs) are downstream components of signal transduction cascade that mediate the molecular responses once cytokinin is perceived by the AHKs receptors (To & Kieber, 2008; Gupta & Rashotte, 2012). To explore whether cytokinin signaling pathway is involved in the regulation of the CMT network after signal is perceived by receptors, we took advantage of the inducible *35S::ARR1ΔDDK-GR* line, in which after application of dexamethasone (DEX) the constitutively active version of the transcription factor ARR1 (ARR1ΔDDK-GR) is translocated to the nucleus, where it regulates transcription of its targets (Sakai *et al*, 2000; Sakai *et al*, 2001). As expected, induction of ARR1ΔDDK by DEX resulted in reduced root growth when compared to either mock-treated transgenic seedlings or wild type exposed to DEX, thus confirming functional experimental set-up (Fig 4A; (Sakai *et al*, 2001)). Monitoring of the CMTs revealed that already 3 h after DEX treatment, ARR1ΔDDK promoted

reorientation of CMTs to oblique orientation in epidermal cells at the TZ and EZ (Fig 4B and C). Furthermore, activation of ARR1ΔDDK-GR reduced sensitivity of root tips to oryzalin-triggered swelling when compared to wild type, and this effect was further accentuated when the ARR1ΔDDK induction was accompanied with external cytokinin treatment (Fig 4D–F). Thus, activation of the cytokinin signaling by ARR1ΔDDK mimics effects of cytokinin on the CTM network.

In addition, we assessed the role of ARR1 in the cytokinin-mediated regulation of CMTs by analyzing the *arr1-3* mutant allele (Fig EV3D and E). Similarly to *ahk4/cre1-12*, in *arr1-3* mutant cytokinin did not promote reorientation of CMTs from transversal to oblique disposition in cell of the EZ (Fig EV3D and E compared to Fig EV1B) as well as treatment with the hormone did not prevent the root swelling triggered by oryzalin and most of root tips of *arr1-3* exhibited severe swelling in the presence of cytokinin (Fig EV3A and C).

Altogether, these results suggest that ARR1 is involved in the transmission of the cytokinin signal to regulate the CMTs in roots.

### Cytokinin interferes with the rapid auxin-driven reorientation of CMTs

Unlike cytokinin, which promoted the reorientation of CMTs from transversal to oblique in epidermal cells of the TZ and the EZ, auxin has been shown to induce a fast reorganization of CMTs from a transversal to a longitudinal orientation in cells at the EZ (Nick *et al*, 1992; Takesue & Shibaoka, 1998; Takahashi *et al*, 2003; Le *et al*, 2005; Vineyard *et al*, 2013; Chen *et al*, 2014; True & Shaw, 2020) (Figs 2E and EV1B; Table EV1). Thus far, the impact of auxin on CMTs in cells of the TZ and the DZ has not been reported. The different configuration and dynamics of CMTs in cells at distinct growth zones motivated us to inspect the sensitivity of the CMTs also in cells at other cytogenesis phases. Notably, the CMT organization did not change after 60 min of auxin treatment in cells at the TZ and the DZ (Figs 2E and G, and EV1B; Table EV1), indicating that CMTs in rapidly elongating cells might be more sensitive to

auxin-driven reorientation than those in other growth phases. Real-time monitoring of the EB1b-GFP reporter revealed that when compared to mock treatment auxin enhanced the growth rates at the plus-end of CMTs in cells of the TZ, but without significant

change in the EZ (Fig 2H and I; compare Fig EV1F to C; Movie EV5). These results suggest that auxin might regulate the CMT network differently from cytokinin. To test the interaction between auxin and cytokinin, we analyzed the response of the CMT network

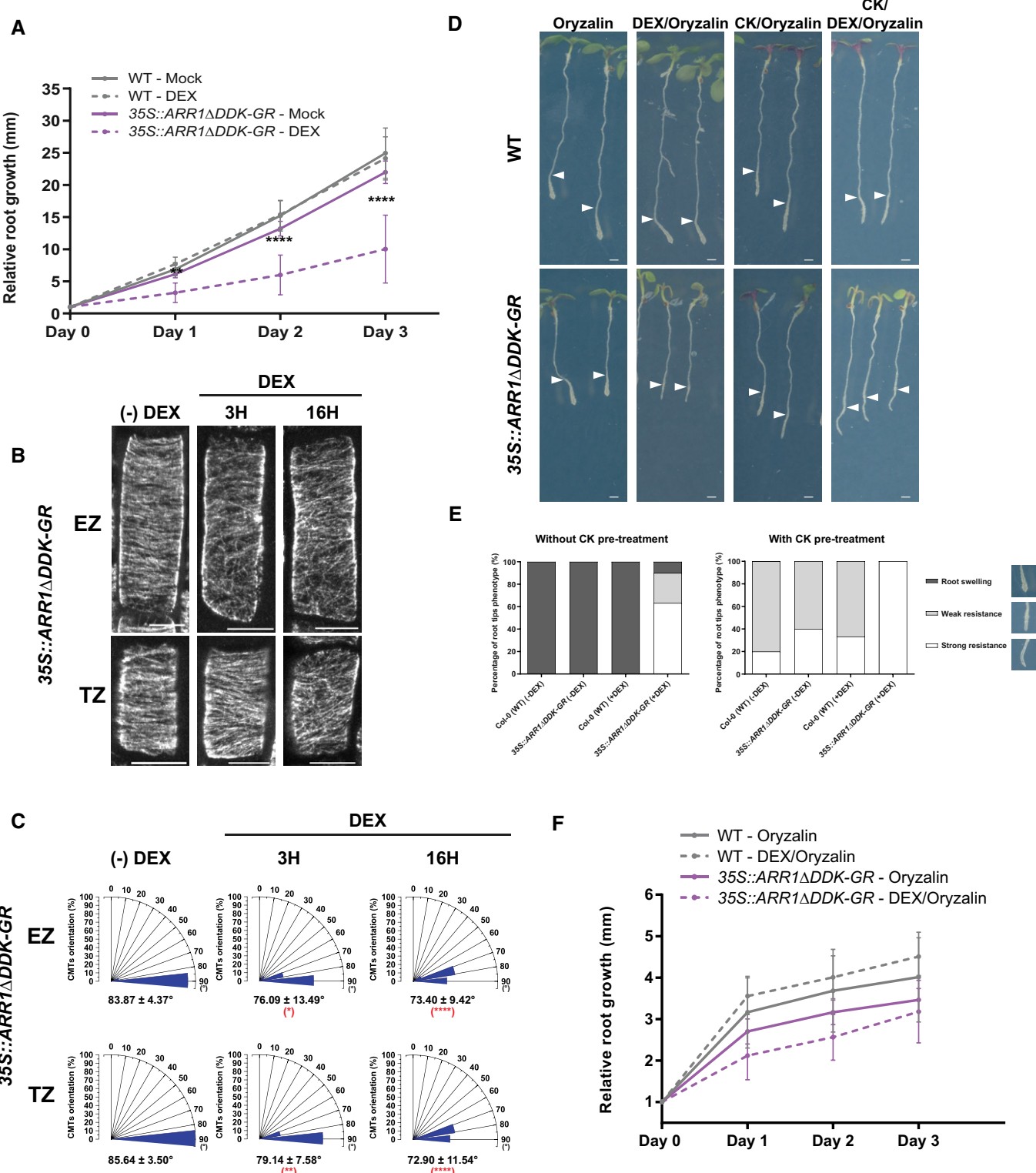

Figure 4.

**Figure 4. The ARR1 cytokinin signaling component is involved in fine-tuning the CMTs network.**

A   Relative root growth of wild type (Col-0) (gray lines) and *35S::ARR1ΔDDK-GR* (purple lines) seedlings grown on mock (DMSO; continuous line) or with dexamethasone (DEX 10 μM; dashed line) supplemented medium. Five-day-old seedlings (Day 0) were monitored for 3 days. Mean ± s.d.; **$P < 0.01$, ****$P < 0.0001$ by Student's *t*-test, referred to wild-type–mock. $n = 12–14$ roots.

B   Immunostaining of α-tubulin in epidermal cells of the transition zone (TZ) and the elongation zone (EZ) of *35S::ARR1ΔDDK-GR* roots after non-treatment, 3 h, or 16 h of treatment with dexamethasone (DEX 10 μM). Scale bar 10 μm.

C   Histograms present the CMT orientation distributions (%) in epidermal cells of the TZ and the EZ treated as indicated in (B). Mean ± s.d. (*$P < 0.05$, **$P < 0.01$, and ****$P < 0.0001$ by Student's *t*-test, referred to same root zone without DEX induction). $n = 16–24$ cells per growth zone in eight–nine roots per condition were analyzed.

D, E   Analysis of root sensitivity to oryzalin in wild type (Col-0) and *35S::ARR1ΔDDK-GR*. Seedlings were grown for 5 days on mock (Murashige and Skoog) medium and then transferred to medium supplemented with 1 μM oryzalin, CK with oryzalin (10 μM BAP and 1 μM oryzalin), with or without DEX 10 μM, for 3 days. For the double CK and oryzalin treatment, seedlings were pretreated with 10 μM BAP for 60 min prior to transfer to medium supplemented with both compounds. Representative images; white arrowheads indicate root length at day of transfer. Scale bar 1 mm (D). Quantifications calculated as percentage (%) of root tips exhibiting swelling (dark gray bars), and weak and strong resistance to oryzalin (gray and white bars, respectively). On the right, representative images of root phenotype categories. $n = 10–25$ roots per treatment were evaluated (E).

F   Relative root growth of wild type (Col-0) (gray lines) and *35S::ARR1ΔDDK-GR* (purple lines) seedlings grown on oryzalin 1 μM (continuous line) or on oryzalin 1 μM and DEX 10 μM (dashed line) supplemented media monitored for 3 days. Day 0, day of transfer. Mean ± s.d.; $n = 12–14$ roots.

to the combined hormonal treatment in the wild type and cytokinin receptor mutants. When auxin and cytokinin were applied simultaneously, CMTs rapidly reoriented from transversal to longitudinal in cells at the EZ similarly as after application of auxin only (Appendix Fig S5A). Intriguingly, pretreatment with cytokinin interfered with a rapid, auxin-driven reorganization of CMTs in epidermal cells of the EZ and promoted the oblique orientation of CMTs (Figs 2F and G and EV1B; Appendix Fig S5B and C; Table EV1). Furthermore, monitoring of the EBb1-GFP reporter revealed that auxin did not interfere with the cytokinin-mediated decrease of the CMT plus-end growth rates (Figs 2H and I, and EV1G and Movie EV6). These results imply that both hormones control CMTs differently and that cytokinin might interfere with the auxin-driven reconfiguration of the CMT network. However, unlike the rapid effects of auxin, cytokinin requires extended pretreatments to counteract the auxin effects on CMTs, hinting at the involvement of additional regulatory steps.

Next, we assessed whether cytokinin receptors and signaling play a role in the cytokinin–auxin regulation of the CMT network. Intriguingly, loss of the AHK4/CRE1 receptor activity interfered with both the impact of auxin and cytokinin on CMTs. Auxin, which in wild-type cells triggered a rapid longitudinal rearrangement of CMTs, promoted only their partial reorientation in *ahk4/cre1-12* (Figs 3C and E versus 2E and G; Appendix Fig S1A versus Fig EV1B; Table EV1). Furthermore, the cytokinin-counteracting effects on the auxin-driven reorientation of CMTs were severely attenuated in *ahk4/cre1-12* and in the loss-of-function *arr1-3* mutant, suggesting that the cytokinin signal perceived by the AHK4 receptor might activate a signaling cascade including the ARR1 transcription factor that interferes with the auxin effects on CMTs (compare Fig 3D and E to Fig 2F and G, Appendix Fig S1A and Fig EV3D and E to Fig EV1B; Table EV1). The auxin-driven reorientation of CMTs was not fully suppressed by cytokinin in either the *ahk2-2* or *ahk3-3* mutants when compared to the wild-type control (Appendix Fig S1B and C compared to Fig EV1B; Appendix Fig S2C and G compared to Fig 2E and G; Appendix Fig S2D and H compared to Fig 2F and G; Table EV1). Hence, the cytokinin signaling mediated through receptor AHK4 that acts partially redundantly with AHK2 and AHK3 might play an important role in fine-tuning the auxin–cytokinin effects on the CMT dynamics.

## The CMT network acquires distinct arrangements in trichoblasts and atrichoblasts correlating with auxin and cytokinin response levels

Noteworthy, two morphologically different epidermal cell types, non-root hair atrichoblast and root hairs forming trichoblasts, exhibited strikingly different cytokinin and auxin activities (Fig EV2A and B). Whereas in atrichoblasts a cytokinin activity was higher than that of auxin, in trichoblasts the auxin activity prevailed (Fig EV2A, B and D). Hence, we used these local differences in the auxin–cytokinin response ratio to monitor CMTs in both epidermal cell types. Unlike the regular oblique arrangement of CMTs in atrichoblasts, longitudinally arranged CMTs were detected in trichoblasts (Fig EV4C, Movie EV9). Furthermore, in atrichoblasts CMTs exhibited a high tolerance to oryzalin-triggered depolymerization, whereas in trichoblasts CMTs were highly sensitive to oryzalin, of which the impact was not prevented effectively by cytokinin (Fig EV4D and E). These results reveal a correlation between the auxin–cytokinin ratio and the CMT dynamics in two morphologically distinct epidermal cell files and are in line with other observations that suggest that cytokinin might modulate the CMT network activity.

## Epidermal cells in different cytogenesis phases exhibit distinct sensitivities to cell expansion inhibition mediated by auxin and cytokinin

Cytokinin and auxin differ in their impact on CMT arrangements and dynamics. Whereas an increase in cytokinin reduces microtubule growth at plus-end and promotes the oblique set-up of CMTs in cells of the TZ and the EZ, auxin-driven reorientation of CMTs in a longitudinal direction is most pronounced in cells of the EZ. To explore whether or how the observed differences in the hormonal regulation of CMTs correlate with the impact of auxin and cytokinin on cell expansion, we analyzed the elongation rate of cells in individual root zones after hormonal treatments. Both hormones were adjusted to concentrations that resulted in similar inhibition of the overall root growth rate (RGR), which was measured in 5-day-old seedlings expressing the plasma membrane marker (*pUB10::EYFP-NPSN12*/W131Y) (Geldner *et al*, 2009) transferred to medium without, cytokinin or auxin supplementation for 1 and 4 h followed by a

real-time imaging for 2 min (Fig 5A and B). RGR and cell expansion were calculated from a plot of two differently color-coded time-lapse images corresponding to 0 and 2 min from the imaging start (Fig 5A and B). One hour after transfer to mock medium, RGR of wild-type roots corresponded to $3.47 \pm 0.17$ µm/min and it remained unchanged also after 4 h ($3.29 \pm 0.11$ µm/min) (Fig 5B). Treatment with either cytokinin or auxin reduced RGR when compared to mock-treated roots. One hour after exposure to cytokinin or auxin, RGR decreased from $3.47 \pm 0.17$ to $1.99 \pm 0.05$ and $2.1 \pm 0.10$ µm/min, respectively, and after 4 h was further reduced to $1.09 \pm 0.06$ and $1.36 \pm 0.14$ µm/min, respectively (Fig 5B). To evaluate the cell elongation profiles along the longitudinal root growth axis, we analyzed the growth rate of epidermal cells at the TZ and the EZ (Fig 5C). As expected, the cellular elongation rate at the EZ ($0.58 \pm 0.05$ µm/min) was approximately 2.5- to 3-fold higher than that of cells at the TZ ($0.22 \pm 0.02$ µm/min; Fig 5D and E). Treatment with cytokinin for 1 h decreased the cell elongation rate in both the TZ and the EZ by 60 and 50%, respectively, and remained unchanged also 4 h after treatment (Fig 5D and E). Intriguingly, auxin applied for 1 h reduced the cell elongation rate in the TZ by only 20%, whereas the growth rate of cells in the EZ was reduced by 60% when compared to that of mock-treated roots (Fig 5D and E). After 4 h of auxin treatment, the elongation rate of cells in the TZ remained reduced by 20%, but that in the EZ decreased by 80% when compared to that of mock-treated roots (Fig 5D and E). These results suggest that although both auxin and cytokinin affected the overall root growth rate almost equally, auxin targets primarily cells at the EZ, whereas cytokinin inhibits expansion of cells at both the TZ and the EZ.

Considering the contribution of cytokinin receptors, in particular of AHK4, to regulation of the CMT activity, we analyzed the AHK4 role in the hormone-regulated cell elongation and overall root growth. RGR of *ahk4/cre1-12* roots on mock medium did not significantly differ from the wild-type control and remained constant after 1 and 4 h of incubation (Fig 5B–E). When exposed to hormones, the sensitivity of RGR of the *ahk4/cre1-12* mutant was significantly lower to both cytokinin and auxin than that of the wild type (Fig 5B). Although the epidermal cells in both the TZ and the EZ exhibited a reduced sensitivity to cytokinin, primarily the attenuated cell growth inhibition in the EZ renders RGR insensitive to auxin in *ahk4/cre1-12* when compared to the wild type (Fig 5D and E).

Altogether, these results hint at considerable differences in auxin and cytokinin sensitivities of cells at different phases of growth and differentiation. Whereas cytokinin seems to constrain cellular expansion during the transition and elongation growth phases, the highest sensitivity to auxin is restricted to a shorter developmental window between the transition and full differentiation phases. In line with the AHK4 receptor function in the regulation of the CMT dynamics by auxin and cytokinin, loss of the AHK4/CRE1 activity attenuated the sensitivity to the cytokinin-mediated inhibition of elongation in both the TZ and the EZ, whereas the reduced sensitivity to auxin occurred specifically in cells at the EZ.

### Cytokinin modulates MT dynamics in animal cells

Microtubules are one of the three major cytoskeletal components in eukaryotic cells and are structurally highly conserved among the different kingdoms (Hashimoto, 2015). Despite certain functional diversification, in both plant and animal cells, MTs are essential for common basic cellular processes, such as proliferation, differentiation, and growth (Wasteneys, 2002; Schaller *et al*, 2014). Interestingly, in animal cells cytokinin has been reported to affect various cellular processes, including cell division and cell differentiation (Griffaut *et al*, 2004; Bifulco *et al*, 2008; Tiedemann *et al*, 2008; Casati *et al*, 2011). To explore whether cytokinin might exert such an effect via the modulation of MTs, we monitored the MT dynamics in animal cells exposed to selected cytokinin derivatives. As model system, leukocytes were used, because they harbor highly dynamic MTs that are involved in many pivotal processes, such as division and locomotion (Vicente-Manzanares & Sánchez-Madrid, 2004; Etienne-Manneville, 2010; Mostowy & Shenoy, 2015; Martín-Cófreces & Sánchez-Madrid, 2018). To visualize the growth of MTs, EB3-mCherry, a mammalian MT plus-end protein orthologue of the plant EB1b marker was used (Akhmanova & Steinmetz, 2008, 2015; Renkawitz *et al*, 2019). In leukocytes, MTs nucleate at the centrosome and grow at their plus-end toward the plasma membrane, thus forming a typical astral array. By automated tracking of EB3-mCherry comets (Applegate *et al*, 2011) (plusTipTracker v1.1.4), the average growth rate of MTs in control cells was approximately 18–19 µm/min (Fig EV5A). Similarly to plants, application of cytokinins, either 6-benzylaminopurine (BAP) or *trans*-zeatin reduced the plus-end growth rate of leukocytes by 30–40% compared to mock conditions (Figs 6A and EV5B and C). Kymograph representations and time projections over 30 s obtained from videos before and after treatment (Movies EV10 and EV11) demonstrated the reduced growth rate at the MT plus-ends upon the cytokinin treatment (Figs 6B and EV5D). Unlike cytokinin, auxin did not significantly affect the MT growth at the plus-ends (Figs 6A and EV5B and C).

To test whether cytokinin affected MT sensitivity to depolymerization drug in animal cells as well, cells were treated with nocodazole in the presence of cytokinins. Similarly to oryzalin, nocodazole inhibits the MT assembly in animal systems, leading to their rapid depolymerization. Treatment of leukocytes with 300 nM nocodazole completely depleted the MTs as early as 10 min post-treatment (Davidse, 1986) (Fig 6C). Interestingly, pretreatment with both cytokinin derivatives (BAP and *trans*-zeatin) prevented the nocodazole-induced depolymerization of the MT cytoskeleton scaffold when compared to the DMSO control (Fig 6D and E). Neither adenine, a molecule structurally similar to cytokinins, nor auxin were able to interfere with the nocodazole effect on MTs (Fig 6D and E). Thus, we hypothesize that the MT cytoskeleton of plant and animal cells might share a partially conserved cytokinin-sensitive control pathway.

## Discussion

Growth of all organisms is a dynamic process driven by key cellular activities, such as capacity to divide, to expand and to differentiate into specific functions. In plants, these elementary cellular functions are under tight hormonal control, with auxin and cytokinin playing dominant roles. A balance between auxin and cytokinin has a decisive morphogenetic function, and as demonstrated by Skoog and Miller (1957), who used plant callus as a model system, a ratio between both hormones provides instructive signals either to

maintain cell proliferation, or to promote root or shoot formation. *In planta*, this balance between auxin and cytokinin determines key tissue patterning and organogenic events, including a root pole specification during embryogenesis, xylem and phloem differentiation in vasculature, lateral root and shoot organogenesis, meristem activity maintenance, or phyllotaxis (Reinhardt *et al*, 2000; Werner *et al*,

2003; Laplaze *et al*, 2007; Müller & Sheen, 2008; Bielach *et al*, 2012; De Rybel *et al*, 2014).

Here, we assessed a role of cytokinin and its interplay with auxin in the regulation of the CMT network during the cellular progress from the proliferative to the differentiation phase in *Arabidopsis* roots. In agreement with previous reports, our real-time analyses of

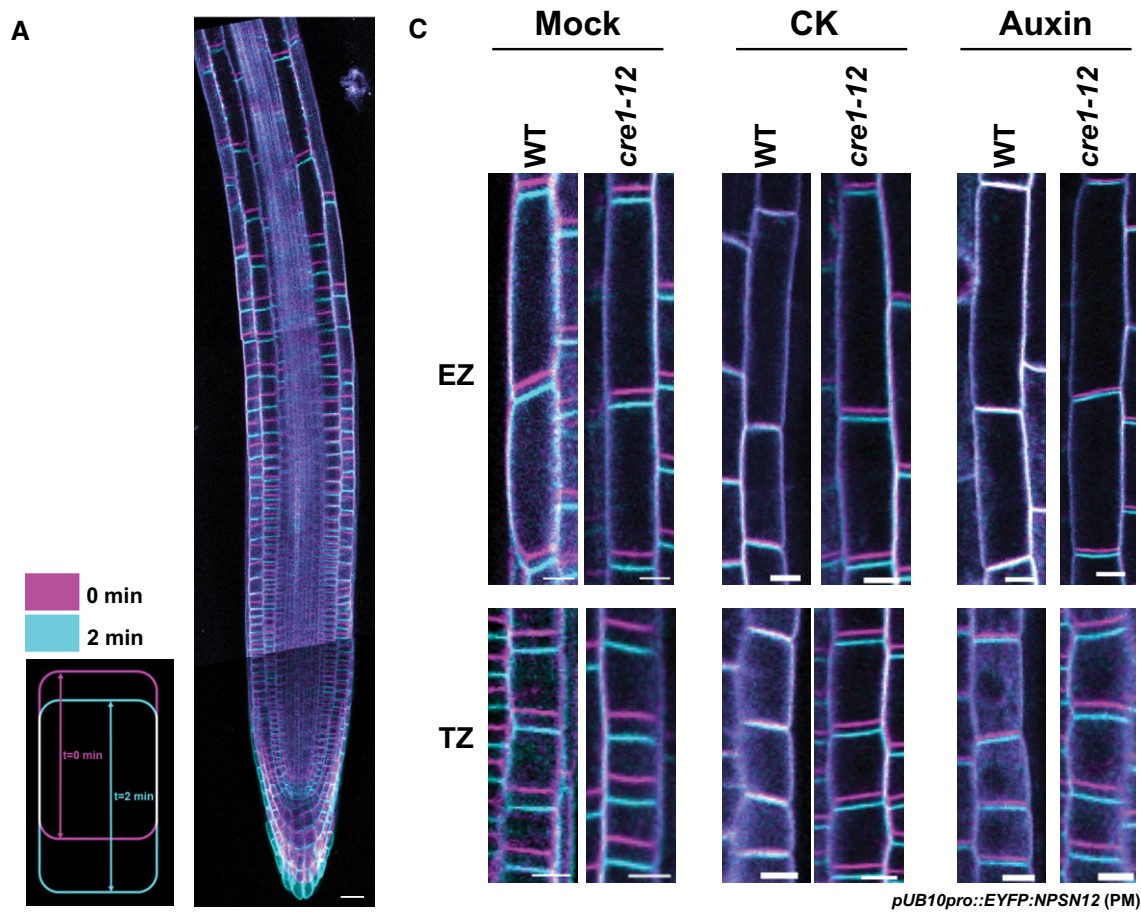

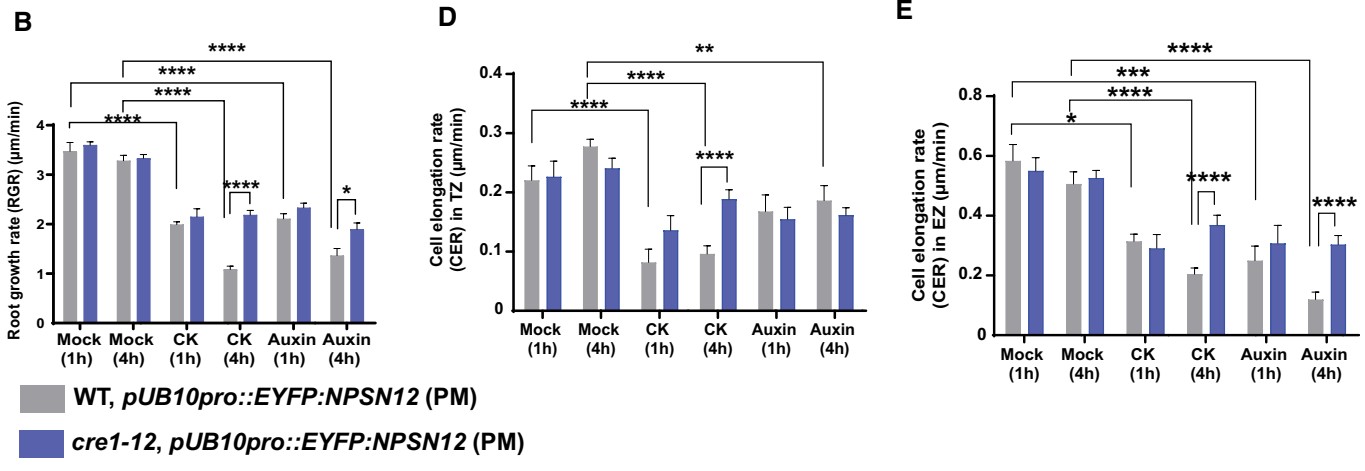

**Figure 5.**

◄

**Figure 5.  Cytokinin and auxin affect distinct root growth zones.**

A    Time-lapse imaging of *Arabidopsis* root expressing the plasma membrane marker (PM) *pUB10::EYFP-NPSN12* (wave line W131Y) (Geldner *et al*, 2009). Root growth recorded for 2 min and overlay of root tip images at time points 0 (magenta) and 2 min (cyan) used for the calculation of the root growth rate (RGR) and the cell elongation rate (CER). Scale bar 25 μm.

B    RGR of wild type (gray bars) and *cre1-12* (blue bars) treated for 1 and 4 h with mock (DMSO), CK (10 μM BAP), or auxin (0.1 μM NAA). Mean ± s.d. (*P < 0.05 and ****P < 0.0001 by Student's *t*-test, n = 5–8 roots per condition in three independent replicates).

C–E  Analysis of CK and auxin effects on CER of epidermal cells. Epidermal cells of the TZ and the EZ in the wild type and *cre1-12* with PMs visualized with the YFP-NPSN12 marker. Overlay of images of epidermal cells at times 0 (magenta) and 2 min (cyan) is presented. Scale bar 10 μm (C). CER (μm/min) of root epidermal cells of the TZ (D) and the EZ (E) of the wild type (gray bars) and *cre1-12* (blue bars) mutant. Hormone concentrations and conditions of treatment (D, E) were as described for (C). Mean ± s.d. (*P < 0.05, **P < 0.01, ***P < 0.001, and ****P < 0.0001 by Student's *t*-test, n = 15–20 cells per root growth zone in three–six roots per condition, in three independent replicates).

epidermal cells in *Arabidopsis* roots showed that CMTs acquire distinct arrangement and dynamics as cells undergo differentiation (Sugimoto *et al*, 2000; Baskin *et al*, 2004; Le *et al*, 2004; Oda, 2015). Transversally arranged CMTs in cells at the TZ and the EZ reorient to an oblique positioning and exhibit a retarded growth at the plus-ends in differentiated cells at the DZ. Different CMT arrangements and dynamics in cells at distinct cytogenesis phases are supported by their sensitivity to depolymerizing drugs. Whereas CMTs in cells at the TZ and the EZ are sensitive to oryzalin, they remain largely intact in the presence of the depolymerizing drug in cells at the DZ. Adaptation of CMT arrangements and dynamics correlate with the need to accommodate different cellular activities, such as demand for effective delivery of new cell wall components during rapid anisotropic growth at the TZ and the EZ (Rasmussen *et al*, 2013; Li *et al*, 2015; Elliott & Shaw, 2018). Accordingly, the different structures and sensitivity of CMTs to depolymerizing drug observed in two morphologically distinct epidermal cells comply with the specific cellular requirements for the formation of root hairs in trichoblasts when compared to non-root hair atrichoblasts (Löfke *et al*, 2015; Salazar-Henao *et al*, 2016; Shibata & Sugimoto, 2019).

The gradual reconfiguration of CMTs in epidermal cells as they pass through distinctive growth areas corresponds with changes in cytokinin and auxin response patterns (Smet *et al*, 2019). The increase in the cytokinin-to-auxin activity in more differentiated cells is accompanied with reduced growth of CMTs at plus-end, their rearrangement from transversal to oblique disposition and reduced sensitivity to oryzalin, pointing at the role of cytokinin in regulation of CMTs. The cytokinin function as a hormonal regulator of the CMT network is further supported by finding that increase of the cytokinin activity (either by exogenously applied hormone or genetically, by activation of the ARR1 signaling pathway component) in cells of the TZ and the EZ triggers changes in disposition and activity of CMTs resembling that detected in differentiated cells. Our data support a role for the signal perception in the cytokinin-mediated regulation of the CMT network with a major function for the AHK3 and AHK4/CRE1 receptors. While loss of the AHK3 activity interferes with CMTs as cells progress from less to more differentiated status, deficiency in the AHK4/CRE1 attenuates the effects of increased levels of cytokinin on the CMT network, such as (i) reorientation of CMTs to an oblique positioning in cells of the TZ and the EZ, (ii) reduction in the growth rate at the CMT plus-ends, or (iii) depolymerization inhibition by oryzalin. We hypothesize that while AHK3 might contribute to fine-tuning of CMTs along the longitudinal root growth axis, AHK4 role might be more prominent under conditions that lead to fluctuations in cytokinin levels (e.g., different type of stresses).

Recently, auxin has been reported to trigger rapid rearrangements of CMTs from a transversal to a longitudinal orientation in epidermal cells at the EZ (Chen *et al*, 2014; True & Shaw, 2020). Here, we show that CMTs in cells that already progressed into the differentiation phase are not responsive to the auxin-driven reorientation, hinting at an alteration of the sensitivity to auxin of CMTs during cytogenesis. We hypothesize that the reduced auxin sensitivity might be the consequence of the cytokinin-driven modification of CMTs in differentiated cells. Our assumption is supported by observation that cytokinin reduces sensitivity of the CMT network to auxin in less differentiated cells of the TZ and the EZ. Accordingly, in the mutants deficient in cytokinin perception and signaling including *ahk4/cre1-12* and *arr1* the cytokinin-counteracting effect on the auxin-driven reorientation of CMTs is dramatically attenuated.

Progress of root epidermal cells from the proliferative to the differentiation phase is accompanied with changes in cell shapes and the profound anisotropic expansion along the growth axis. Inevitably, CMTs play a role in maintaining the growth anisotropy and as well provide a framework for the delivery of components that strengthen cell walls of expanding cells (Hashimoto, 2015; Oda, 2015; Elliott & Shaw, 2018). Depletion of CMTs results in the typical swelling of cells at the root tip that can be prevented by cytokinin in an AHK4-dependent manner, thus supporting cytokinin as a regulatory factor of the CMTs that contributes to the cell shape maintenance during cellular growth.

A relation between CMT rearrangements and cellular elongation is still a matter of debate. Typically, rapidly elongating cells exhibit transversal CMT arrangements, optimal for the efficient delivery of cell wall components to the cell wall, an oblique orientation of CMTs is characteristic of differentiated cells, whereas the longitudinal one correlates with the rapid cell expansion inhibition in response to auxin (Chen *et al*, 2014). Although a simple quantification of the cell growth rate is not sufficient to dissect the specificities of the regulation of CMTs by auxin and cytokinin in relation to the cell growth, we found that at the concentrations with a comparable impact on the overall root growth rate, cells of the TZ exhibited a higher sensitivity to cytokinin than to the auxin-mediated inhibition. Unlike for cells at the TZ, the growth suppression of cells at the EZ is higher by auxin than by cytokinin, suggesting that root growth inhibition by auxin and cytokinin might involve distinct mechanisms.

Microtubules are part of the cytoskeleton with largely conserved structures and functions across different species (Hashimoto, 2015). Here, we show that the MT network in animal cells exhibits a pronounced cytokinin sensitivity. In leukocytes, cytokinin

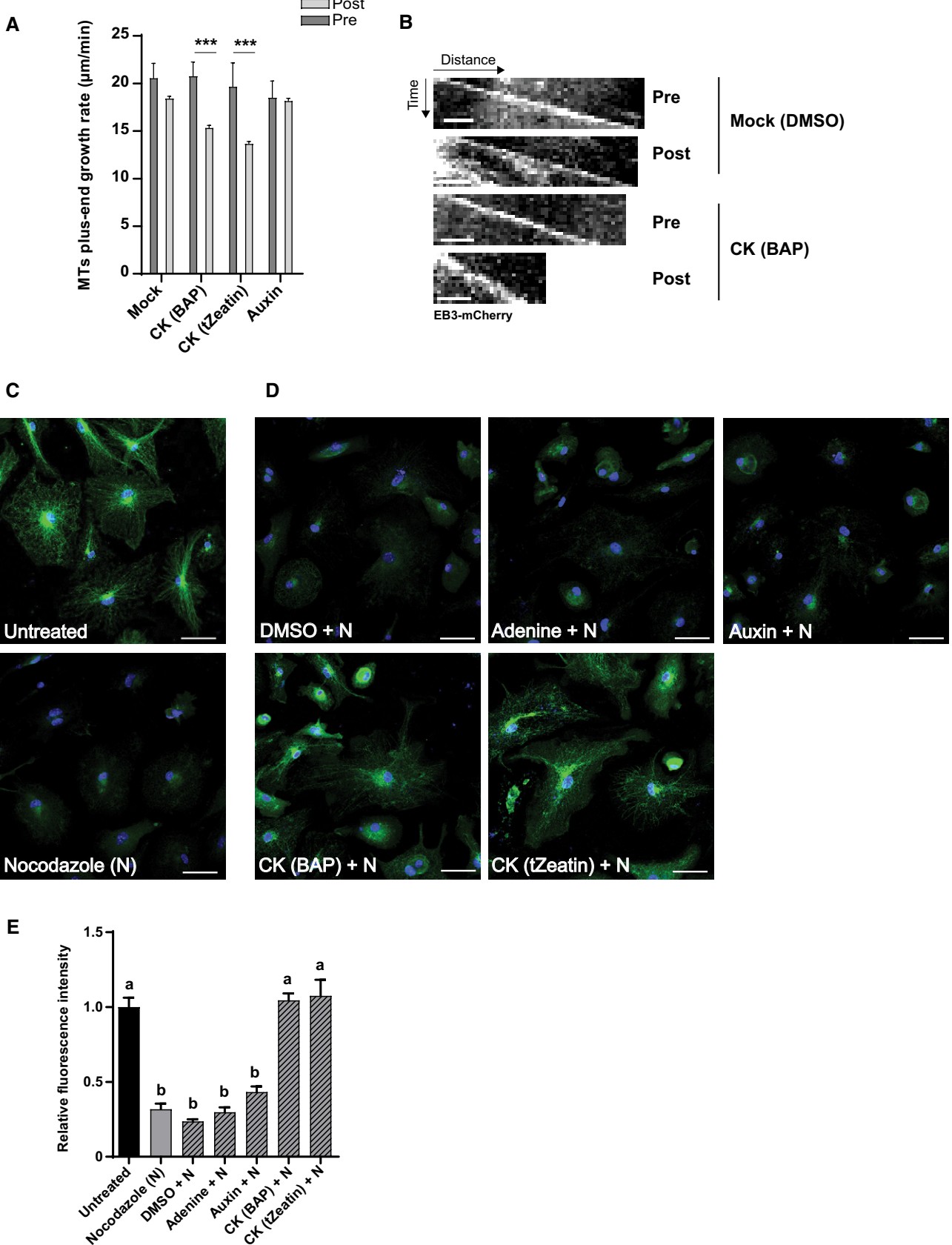

**Figure 6.**

**Figure 6. Cytokinin modulates MT dynamics in animal cells.**

A MT plus-end growth rates (μm/min) in leukocytes expressing the EB3-mCherry plus-end marker before (dark gray) and after (light gray) treatment with mock (DMSO), CK (10 μM BAP or 10 μM *trans*-zeatin), and auxin (0.1 μM NAA). MT plus-end growth rates were monitored for 10 min before and for 10 min after each treatment. The number of MT plus-end events (EB3-mCherry) tracked per cell was 100–200/min. Mean ± s.d. (***$P < 0.001$ by Student's *t*-test, $n = 1$–3 cells per condition in four independent replicates).

B Kymograph representation of individual EB3-mCherry comets tracked for 30 s in leukocytes cells, before and after treatment with mock (DMSO) or CK (10 μM BAP). Scale bar 2 μm.

C–E Immunostaining of α-tubulin (green) in leukocytes mounted in media containing DAPI (blue, nuclear marker) and quantification of immunolabeled MTs. Cells were not or treated with 300 nM nocodazole (N) for 10 min (C). Leukocytes pretreated for 30 min with mock (DMSO), CK (10 μM BAP or 10 μM *trans*-zeatin), adenine (10 μM), or auxin (10 μM NAA) prior to a 10-min treatment with 300 nM nocodazole (D). Scale bar 50 μm (C, D). (E) Relative fluorescence intensity of MTs in leukocytes treated with nocodazole, or nocodazole and hormones as described for (D) compared to untreated (black bar) cells. Mean ± s.d. ($P < 0.05$ by two-way ANOVA, $n = 18$–33 cells per condition in three independent replicates).

decreases the MT plus-end growth rates and antagonizes the effects of depolymerizing drugs, implying that similarly as in plant cells also in animal cells, cytokinin might regulate activity of the MT network. Unlike cytokinin, auxin does not interfere with MTs in animal cells, indicating that the auxin-mediated regulation of MTs might be specific to plants, whereas cytokinin presumably targets a more general activity-controlling mechanism of the MT network. However, how animal cells might perceive and transduce the cytokinin signal to regulate MTs is fully unknown. Thus far, no hints exist on a two-component phosphorelay or on any orthologous pathway that could transduce the cytokinin signal in animal cells (Attwood, 2013). It can be speculated that in animal cells cytokinin might "promiscuously" activate pathways that control the MT activity, which in plant cells is targeted by the histidine kinase-mediated signaling. Noteworthy, several cytokinin derivatives, including kinetin, *trans*-zeatin, BAP, and $N^6$-($\Delta^2$-isopentenyl) adenine, display considerable biological activities also in animal cells. Cytokinin effects have been described on cell proliferative activities or cell differentiation in some cancer diseases that potentially can be mechanistically linked with the regulation of the MT activity (Ishii *et al*, 2003; Griffaut *et al*, 2004; Bifulco *et al*, 2008; Lee *et al*, 2012; Jabłońska-Trypuć *et al*, 2016; Othman *et al*, 2016; Hönig *et al*, 2018; Kadlecová *et al*, 2019). Thus, the identification and characterization of the cytokinin perceptive pathway that controls the MT activities is an exciting area of research that might potentially reveal intriguing aspects of the evolutionary conservation of core cell-regulatory systems.

## Materials and Methods

### Plant material

*Arabidopsis thaliana* (L.) Heynh. plants were used in this work. The transgenic lines have been described elsewhere: *35S::MAP4-GFP* (Marc *et al*, 1998), *EB1b::EB1b-GFP* (Dixit *et al*, 2006), *cre1-12* (Higuchi *et al*, 2004), *ahk2-2* (Higuchi *et al*, 2004), *ahk3-3* (Higuchi *et al*, 2004), *arr1-3* (Argyros *et al*, 2008), *35S::ARR1ΔDDK-GR* (Sakai *et al*, 2001), *pTCSn::ntdTomato:tNOS-pDR5v2::3nGFP* (Smet *et al*, 2019), *pUB10::EYFP-NPSN12* (W131Y/PM) (Geldner *et al*, 2009), and *35S::mCherry-TUA5* (Gutierrez *et al*, 2009). The lines *cre1-12*, *35S::MAP4-GFP*; *ahk2-2*, *35S::MAP4-GFP*; *ahk3-3*, *35S::MAP4-GFP*; *cre1-12*, *EB1b-GFP*; *cre1-12*, *W131Y*; *ahk2-2*, *W131Y*; and *ahk3-3*, *W131Y* were generated by crosses, and homozygous lines for both mutant allele and reporter were selected.

### Growth conditions

Surface-sterilized seeds were plated on half-strength (0.5×) Murashige and Skoog (MS) medium (Duchefa) with 1% (w/v) sucrose and 1% (w/v) agar (pH 5.7). Seeds were stratified for 2–3 days at 4°C in the dark. Seedlings were grown on vertically oriented plates in growth chambers at 21°C under long-day conditions (16-h light/ 8-h dark photoperiod) and white light (W), provided by blue and red LEDs (70–100 μmol/m$^2$/s of photosynthetically active radiation), if not stated otherwise.

### Animal cells (Leukocytes)

Leukocytes (dendritic cells and macrophages) were grown and maintained at 37°C in a humidified incubator with 5% CO$_2$ and routinely tested for mycoplasma contamination. Cells were differentiated from immortalized hematopoietic precursors of bone marrow cells, which had been initially extracted from femur and tibia of 6- to 8-week-old C57BL/6J wild-type mice. EB3-Cherry in immortalized hematopoietic precursors was expressed stably by lentiviral infection of precursor cells (preprint: Kopf *et al*, 2019). In brief, hematopoietic precursors were differentiated in R10 culture medium (RPMI 1640 supplemented with 10% fetal calf serum (FCS), 2 mM L-glutamine, 100 U/ml penicillin, 100 mg/ml streptomycin, and 0.1 mM 2-mercaptoethanol (all Gibco) supplied either with the granulocyte-macrophage colony-stimulating factor (GM-CSF) hybridoma supernatant for dendritic cell (DC) differentiation or the L292 supernatant for macrophage differentiation. Fresh medium was added on the differentiation days 3 and 6. For subsequent assays, cells were used between the differentiation days 8 and 10, and DCs were stimulated with 200 ng/ml lipopolysaccharide (LPS) (*Escherichia coli* 0127:B8 Sigma) to induce maturation.

### Pharmacological and hormonal treatments

#### *Arabidopsis*

Seedlings 4–5 days old were transferred onto solid MS medium with or without the indicated chemicals. The drugs and hormones used were as follows: 6-benzylaminopurine (BAP; 0.1 μM and 10 μM) and *trans*-zeatin (10 μM) as cytokinin sources, 1-naphthaleneacetic acid (NAA; 0.1 μM) as an auxin source, and oryzalin (1 μM) as a microtubule-depolymerizing drug. Mock treatments contained equal amounts of solvent (DMSO). For combined treatments (either cytokinin and auxin, or cytokinin and oryzalin), pretreatment with cytokinin for either 30 or 60 min was followed by a 60-min concomitant

treatment with both chemicals at the concentrations specified. Dexamethasone (DEX) 10 μM applied for 3 and 16 h was used to activate *35S::ARR1ΔDDK-GR* line. Propidium iodide (PI) (10 μM) was used for staining of the cell walls.

### Animal cells

Leukocytes were treated with the R10 culture medium prewarmed to 37°C supplemented with hormones at the specified concentrations. Hormones and drugs used were as follows: 10 μM BAP and 10 μM *trans*-zeatin as active cytokinins; adenine (10 μM) as an inactive cytokinin metabolization product; 10 μM NAA as an auxin source; and 300 nM nocodazole as a MT-depolymerizing agent. Mock treatments contained equal amounts of solvent (DMSO). For each treatment, a double concentration of the drug/hormone (2×) was resuspended in the complete medium and cells were added in a 1:1 volume proportion to achieve a final drug/hormone concentration of 1×.

### Imaging

Confocal images were taken with LSM 700 and LSM 800 vertical-stage laser scanning confocal microscopes (Zeiss) (von Wangenheim *et al*, 2017) equipped with a 20×/0.8 Plan-Apochromat M27 objective; and a LSM 800 inverted confocal scanning microscope (Zeiss) equipped with a 40× Plan-Apochromat water immersion objective. Fluorescence markers were excited at 488 nm (GFP, YFP and Alexa Fluor 488), 555/561 nm (mCherry), and 561 nm (tdTomato and PI). Emission was collected through band-pass filters: 490–530 nm (GFP, YFP, and Alexa Fluor 488), 570–700 nm (mCherry), 560–700 nm (tdTomato), and 617–700 nm (PI). For a vertical root tracking, 1 × 3 tiles, six *z*-stacks (1 μm), and time lapse (one picture/10 min) were used. For the CMT quantification of the angle orientations and fluorescence signals, a maximum intensity projection of confocal stacks including exclusively the cell cortical area was utilized. Images were post-processed, profiles were measured, and colocalization analyzed with the Zeiss Zen 2011 program, ImageJ (National Institute of Health, http://rsb.info.nih.gov/ij), Adobe Illustrator CC 2018, GraphPad Prism 8, and Microsoft PowerPoint programs.

### Quantification of MT orientation

FibrilTool plug-in of ImageJ was used (Boudaoud *et al*, 2014) to quantify the CMT orientation in individual cells. Orientation was measured as an angle between CMTs and the longitudinal root growth axis, with 0°, 45°, and 90° corresponding to the longitudinal, oblique, and transversal orientations, respectively, and was scored in 10–21 cells per root growth zone, and 5–9 roots per condition were analyzed in 3–4 independent replicates. The number of samples is further specified in the figure legends. Histograms of MT orientations were obtained with Origin, Version 2016 (OriginLab Corporation, Northampton, MA, USA.).

### MT plus-end growth rate measurements

#### Arabidopsis roots

For MT plus-end protein (EB1b-GFP) detection and tracking, an Andor spinning disk microscopy (CSU X-1, camera iXon 897 [back-thinned EMCCD], FRAPPA unit and motorized piezo stage) was used. The EB1b-GFP trajectories were observed by the spinning disk confocal system with a 63× water immersion objective. Videos were acquired with a 500-ms exposure time, five *z*-stack 0.7 μm/each, every 3.2 s, for 20–30 min. The settings of excitation and detection were as follows: 488 nm for GFP, 505–550 nm. All images in a single experiment were captured with the same settings.

EB1b-GFP spots were tracked by single-particle tracking (Track-Mate) (Tinevez *et al*, 2017), and the tracks were further analyzed with a custom script. Low-quality short tracks were discarded, and the remaining tracks were corrected for sample drift. At each time point, the frame-to-frame displacements for all detected spot were averaged. The resulting drift vector of the sample is then smoothed using a moving average filter with a span of 10 time frames. This sample drift vector was then subtracted from the spot motion, and the mean speed of all valid tracks was calculated. The mean speed was then binned into 1-min intervals. EB1b-GFP trajectories were evaluated in videos of individual epidermal cells; therefore, the results are not influenced by the movement of the rest of the root (furthermore, the rest of the root is not taken into account for the stabilization). Data from $n = 3$ biological replicates per hormonal treatment were pooled (Figs 1D, 2I and 3G). The number of samples is further specified in the figure legends.

The TrackMate tool allows the plotting of the speed velocities of individual dot tracks (MT plus-ends, EB1b-GFP) with a color code (dark blue and dark red, slow and fast speed, respectively). The maximum intensity projections of 30-s imaging (for the different mutants and treatments) were used to visualize the different MT plus-end growth speeds. Manual tracking of single MT plus-end proteins (EB1b-GFP) was performed using the ImageJ software (NIH) and the Multi Kymograph tool to evaluate the plus-end growth rates.

For analyzing the CMT growth using the *35S::mCherry-TUA5* marker, an Andor Dragonfly spinning disk (Andor Zyla 4.2 Megapixel sCMOS camera, Nikon Ti2 inverted microscope) microscope was used. The mCherry-TUA5 signal was observed by the spinning disk confocal system with a CFI P-Apo 60× Lambda/NA 1.40/WD 0.13 mm oil immersion objective and using 40 μm pinhole. Videos were acquired with an 800-ms exposure time, every 3 s, for 5 min. Excitation with 561 nm laser was used for mCherry fluorescence marker. All images in a single experiment were captured with the same settings to record individual epidermal cells of the EZ. Kymographs were constructed drawing a 20 μm line and using the Multi Kymograph tool (ImageJ software, NIH). Kymographs were analyzed using KymoButler tool (Jakobs *et al*, 2019). Data from $n = 3$ biological replicates (with trajectories analyzed $n = 127$ for mock and 118 for CK) per hormonal treatment were pooled (Fig EV1H).

### Animal cells

For MT plus-end protein (EB3-mCherry) detection and tracking, the Andor spinning disk microscopy (CSU X-1, camera iXon 897 [back-thinned EMCCD], FRAPPA unit and motorized piezo stage) was used. The EB3-mCherry trajectories were recorded over 10 min before and after the drug/mock application with a 63× water immersion objective. Videos were acquired with 500-ms exposure time, one single *z*-stack, every 2 s for one individual cell. The MT growth rates (μm/min) were quantified by automated tracking of the EB3-

mCherry comets by means of the plusTipTracker v1.1.4 (Applegate *et al*, 2011). All images in a single experiment were captured with the same settings. Graphs show averages of MT growth rates (Figs 6A and EV5A) and MT plus-end growth speed distributions (histogram representation with Gaussian normalization, Fig EV5C) were analyzed, with $n = 4$ biological replicates per treatment and 1–3 cells per biological replicate.

### Quantification of root and cell elongation rates

For the evaluation of the cell elongation rate, roots of 5-day-old *Arabidopsis* seedlings (1 and 4 h after mock or hormone treatments) were monitored for 2 min, and one picture/30 s was recorded as a single *z*-stack image with the LSM700/800 confocal microscope. Individual epidermal cells were visualized with the plasma membrane marker *pUB10::EYFP-NPSN12* (W131Y). The root growth rate (RGR in μm/min) was calculated based on a shift in the quiescent center (QC) per min and the cell elongation rate (CER in μm/min) on a relative increase in cell length (with the middle of the cell as a reference) that occurred within a 2-min time interval. For RGR, a minimum of five roots per condition was analyzed and for CER, more than 10 cells per root growth zone in three to six roots per treatment in three independent replicates. The number of samples is further specified in the figure legends.

### Expression analysis of auxin and cytokinin reporters in *Arabidopsis* roots

The expression of the dual reporter sensitive to auxin and cytokinin *pTCSn::ntdTomato:tNOS-pDR5v2::3nGFP* (Smet *et al*, 2019) was monitored with the LSM800 confocal vertical microscope in the epidermal cells along the longitudinal root growth axis. The relative fluorescence intensity of the dTomato and GFP reporters in the nuclei of individual epidermal cells of an individual root ($n = 18$ roots) was measured and plotted in the graph using as a reference always the cell 1 at the meristematic zone (Fig 2B).

### Evaluation of MTs and root sensitivities to drug-induced depolymerization

#### *Arabidopsis* roots

Sensitivity of CMTs to depolymerization induced by oryzalin was evaluated in root epidermal cells of 5-day-old seedlings expressing the MAP4-GFP reporter after treatments as described above. To examine the sensitivity to oryzalin of the CMT network, a ratio was calculated between the mean fluorescent intensity measured in a region of interest (ROI) in individual cells, with maximum projections using *z*-stacks that includes only the cortical area (proximal to the plasma membrane), at times 0 and 60 min after the oryzalin treatment. ROIs were manually selected in epidermal cells to exclude cell walls. Per root growth zone, 3–10 cells per root, and 5–8 roots per condition were analyzed in four 4 independent replicates.

Sensitivity to oryzalin of the *Arabidopsis* primary root and the impact of cytokinin on the root were examined in 5-day-old seedlings transferred on solid MS media supplemented with 1 μM oryzalin, or 1 μM oryzalin and 10 μM BAP. The root growth was monitored with a vertically positioned scanner (EPSON perfection

v800 Photo). The root growth rate was calculated (5 days after germination [DAG]) of 5-day-old seedlings ($n = 10$). For the relative root growth, the root length of 5-DAG seedling (day 0 in new treatment-containing media) transferred onto solid MS medium with or without the indicated chemicals was considered as 1. The statistical significance was evaluated by Student's *t*-test. The swelling of root tips was imaged 3 days after transfer with the LSM800 confocal microscope.

#### Animal cells

Sensitivity of MTs to nocodazole-triggered depolymerization was evaluated in leukocytes. Cells pretreated for 30 min with mock or hormone-supplemented media (see above) were incubated in either mock medium or medium supplemented with 300 nM nocodazole for 10 min. After cell fixation, MTs were immunostained with the α-tubulin antibody (Clone YL1/2, AbD Serotec; see below). To evaluate the MT sensitivity to depolymerization, a ratio between a mean fluorescent intensity of the α-tubulin signal in nocodazole-treated or untreated cells was calculated, with $n = 18$–33 cells per condition, in three independent replicates. The number of samples is further specified in the figure legends.

### Immunodetection of α-tubulin

#### *Arabidopsis*

Four-day-old roots were immunostained as previously described (Pasternak *et al*, 2015). Chemicals were fixed with 2% para-formaldehyde (PFA) in modified tryptone soy broth (MTSB; 3 × 10 min in vacuum) supplemented with 0.1% Triton X-100, followed by the hydrophilization with 100% MeOH (65°C, 10 min), 40 min of cell wall digestion at 37°C with 0.2% driselase and 0.15% macerozyme in 2 mM 2-ethane sulfonic acid (MES), pH 5.0, and permeabilization with 3% NP-40, 10% DMSO in MTSB for 20 min at 37°C. Immunostaining for 2 h with a 1:100 dilution of the anti-α-tubulin YOL1/34 rat monoclonal IgG2a (sc-53030, Santa Cruz Biotechnology) and 1:500 dilution of Alexa Fluor 488 goat anti-rat IgG H+L (Thermo Fischer Scientific) were used as primary and secondary antibodies, respectively, with $n = 5$ cells per root growth zone per root per condition and five to eight roots analyzed per condition. More details on the exact number of samples are given in the figure legends.

#### Animal cells

For the immunodetection of α-tubulin, leukocytes (after the respective pharmacological and hormonal treatments, see above) were washed three times with phosphor buffered saline (PBS) and fixed by addition of prewarmed (37°C) 4% PFA. Cells were permeabilized for 15 min with 0.2% Triton X-100 in PBS and washed 3× for 10 min in PBS. Samples were blocked to prevent unspecific binding by incubation for 60 min in 5% bovine serum albumin (BSA) and immunostained for 2 h with primary rat monoclonal anti-α-tubulin (Clone YL1/2, AbD Serotec). After the cells had been washed 3× for 10 min in PBS, they were incubated for 30 min in Alexa Fluor 488-AffiniPure F(ab′)2 (Jackson Immuno) secondary antibody and washed at least 3× with PBS for 5 min. Samples were mounted in non-hardening mounting medium with DAPI 514 (Vector Laboratories) and stored at 4°C in the dark.

## Statistics

The statistical significance was evaluated either with Student's *t*-test (*$P < 0.05$, **$P < 0.01$, ***$P < 0.001$, and ****$P < 0.0001$) or two-way ANOVA. In the boxplots, the center lines show the medians; the box limits indicate the 25th and 75th percentiles as determined by the GraphPad software; whiskers span the minimal to maximal values; and individual data points are represented by dots.

# Data availability

This study includes no data deposited in external repositories.

**Expanded View** for this article is available online.

## Acknowledgements
We thank Takashi Aoyama, David Alabadi, and Bert De Rybel for sharing material, Jiří Friml, Maciek Adamowski, and Katerina Schwarzerová for inspiring discussions, and Martine De Cock for help in preparing the manuscript. This research was supported by the Scientific Service Units (SSUs) of IST Austria through resources provided by the Bioimaging Facility (BIF), especially to Robert Hauschild; and the Life Science Facility (LSF). J.C.M. is the recipient of a EMBO Long-Term Fellowship (ALTF number 710-2016). This work was supported with MEYS CR, project no.CZ.02.1.01/0.0/0.0/16_019/0000738 to J.P., and by the Austrian Science Fund (FWF01_I1774S) to E.B.

## Author contributions
Conceptualization: JCM, AA, JP, MS, EB; Methodology: JCM, AA, AK, AJ-G, KÖ; Formal analysis: JCM, AK, AJ-G, EB; Investigation: JCM, AA, JP, AK, AJ-G, MS, EB; Resources: JP, MS, EB; Writing—original draft: JCM, JP, EB; Supervision: EB; Project administration: EB; Funding acquisition: JCM, MS, EB.

## Conflict of interest
The authors declare that they have no conflict of interest.

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
