## [Review Process File · The EMBO Journal]

Phytohormone cytokinin guides microtubule dynamics during cell progression from proliferative to differentiated stage

Juan Carlos Montesinos, Anas Abuzeineh, Aglaja Kopf, Alba Juanes-García, Krisztina Otvos, Jan Petrášek, Michael Sixt, and Eva Benková

DOI: N/A

Corresponding author: Eva Benková (Eva.BENKOVA@ist.ac.at)

Review Timeline:

Submission Date:	10th Dec 19
Editorial Decision:	24th Jan 20
Revision Received:	5th May 20
Editorial Decision:	9th Jun 20
Revision Received:	12th Jun 20
Accepted:	22nd Jun 20

Editor: Ieva Gailite

Transaction Report:

Thank you for submitting your manuscript for consideration by the EMBO Journal. I apologise for the protracted review process due to delays in review submission over the holiday period. We have now received three referee reports on your manuscript, which are included below for your information.

As you will see from the comments, all reviewers appreciate the presented characterisation of the role of cytokinin in regulation of microtubule dynamics. However, they also raise a number of concerns regarding core aspects of the study that would need to be addressed before they can support publication here. In particular, reviewers indicate open questions regarding the molecular details of cytokinin-mediated regulation of MT dynamics, cytokinin/auxin interplay in this process and the endogenous relevance of this pathway for root growth, as well as issues with technical aspects of the analysis indicated by reviewer #2.

Based on the overall interest expressed in the reports, I would like to invite you to submit a revised version of your manuscript, in which you address the comments of all reviewers. I should add that it is The EMBO Journal policy to allow only a single major round of revision and that it is therefore important to resolve the main concerns at this stage. Please contact me if you would like to discuss feasibility of any of the requested experiments.

Please feel free to contact me if have any further questions regarding the revision. Thank you for the opportunity to consider your work for publication. I look forward to your revision.

Referee #1:

In this exciting manuscript Montesinos and colleagues study the effects of cytokinin on microtubule reorientation during root development. Utilizing the epidermal cells as model system, they first describe the orientation of MTs in the different developmental zone of the root. They show that in the TZ and EZ MTs are arranged transversally, whereas in the EZ obliquely. Subsequently they show that 1 h treatment of cytokinin is sufficient to promote MTs reorientation from transversal to obliquous. They also show that cytokinin dependent MTs regulation is largely reliant on the activity of the AHK4 cytokinin. They demonstrate that cytokinin counteracts the rapid auxin-induced MT re-arrangement in an AHK4 dependent manner at the TZ and DZ. Intriguingly they also show that atrichoblast cells have higher cytokinin activity and obliquous MTs whereas trichoblast cells show higher auxin activity and randomly arranged MTs, supporting their findings.

Finally, utilizing leukocytes as model animal system, authors demonstrate that cytokinin activity on MT arrangement is conserved in animal cells.

Major points:

To this reviewer the study is a fundamental piece in cytokinin research; nonetheless there are some concerns to be addressed prior publication.

My major concern regards the requirement of AHK4 but not of AHK3 in the regulation of MT arrangement. Indeed, AHK4/CRE1 is a cytokinin receptor expressed in vasculature cells, whereas AHK3 is expressed also in epidermal cells. Authors show that cytokinin acts on epidermal cells in an AHK4 dependent manner. Is the activity of cytokinin on MT arrangement cell non autonomous? Please discuss.

Moreover, in case of cytokinin cell non autonomous mechanism on MT arrangement, would it be possible that cytokinin causes rearrangement of MTs via auxin? If so, how could the dynamics of cytokinin and auxin activity on MT be explained?

Also, given that cytokinin promotes MT rearrangement in leukocytes and given that orthologues of two component systems are missing in animals, it would be plausible that cytokinin acts via mechanisms independent by the ARR. Analysis of MT arrangement after inductions of phospho-insensitive overexpressor of ARR1 (35S::ARR1DDK:GR) would be enough to demonstrate this point. It is not clear to me the contribution given by the cytokinin dependent orientation of MT to root development. To test the activity in root development authors reports that plants treated with cytokinin and oryzalin for 72 hours show dampened root growth inhibition effects than plant treated with only oryzalin, however I do not consider this data sufficient to demonstrate the role of cytokinin dependent MT reorientation in development. It is known that cytokinin treatments inhibit root growth after 12 hours of treatment and that in plants exposed to transient cytokinin treatments (transferring plants after sufficient time for root inhibition from media supplemented with cytokinin to media without cytokinin), root restart growing after few hours of transfer. Knowing whether the re-establishment of growth after cytokinin fluctuations is preceded by a re-orientation of MT from obliquus to transversal at the TZ and DZ would represent permit to better understand the developmental importance of the cytokinin mediated MT rearrangement.

Minor comments:

I believe that the manuscript would benefit from a graphical model reporting the activity of auxin and cytokinin on MT arrangement in the different zones of the root.

It is not clear from the text how the TCS::GFP gradient was demonstrated. To this reviewer it looks like the considered gradiend only refer to the one in the TZ, DZ and EZ of the epidermis. Please specify.

Referee #2:

How signaling pathways that control plant growth interface with the core machinery of growth, such as the cytoskeletal, secretory and cell wall biosynthetic machinery, is one of the major frontiers in our understanding of plant growth and function. In organisms with rigid cell walls, including higher plants, a cortical cytoskeleton guides cell wall biosynthesis and cell wall expansion, shaping the growth and morphogenesis of the organism. Here, the authors investigate the relationship of cytokinin signaling with the behavior and organization of the microtubule cytoskeleton in roots of *Arabidopsis thaliana*. An increasing gradient of cytokinin signaling activity has been described from the root meristem to the zone where cells derived from the meristem finally mature and differentiate. The authors describe a correlation of cortical microtubule organization and growth rates of microtubule plus end with this gradient of cytokinin activity (the organizational gradient has been described in previous studies). They then test causation by exogenous application of cytokinin and by genetic disruption of cytokinin receptors. In addition, they investigate possible interactions of the auxin and cytokinin biosynthetic pathways in modulating cytoskeletal behavior and they demonstrate that exogenous application of cytokinin can also perturb plus end growth in animal tissue culture cells. Major conclusions are that cytokinin tunes cytoskeletal organization and behavior across along the root axis, that auxin and cytokinin may to affect cytoskeletal behavior and cell growth by different but interacting pathways, and that plant and animal cells may share a conserved mechanism of action in modulating microtubule growth. The manuscript is lengthy but well organized, describing a large number of experiments. In general, a strength of the study is a quantitative approach to the questions being addressed. It is excellent that a vertically oriented confocal instrument was used to image root and cellular behavior, avoiding

confounding hormonal signaling with gravitropic signaling. This was impressive and is seldom done. The manipulation of cytokinin activity by use of mutants in cytokinin receptors was also a strength, being an important complement to the use of exogenous hormone application, which may not always probe normal physiology well. On the other hand, most major effects of cytokinin application on the cytoskeleton were observed with or depended on exogenous hormone application, including the consequences of receptor mutation. Defects in cytoskeletal organization and plus end growth rates were not observed when receptor function only was perturbed. In addition I had a number of other comments and concerns about the experiments and their interpretation, as presented below.

1. A major part of the study is the measurement of microtubule growth rates as measured by EB1-GFP labeling. A systematic analysis of plus end microtubule growth in the plant root growth zone has not previously been reported. One issue with expression of EB1 which is not mentioned, but should be discussed in the text, is that EB1 overexpression changes microtubule dynamics - speeding up growth rates and stabilizing the growth phase. In the context of this study, it would be good to know if the observed pattern of plus end growth rates along the root axis is sensitive to EB1 expression itself. The possible influence of EB1 expression on patterns of plus end growth rates might be assessed by performing kymograph analysis on cells expressing fluorescent protein (FP) fusions to tubulin (using suitably registered datasets). Use of FP-tubulin fusions is preferable to using lines expressing MAP4-GFP as expression of MAP4-GFP also has significant effects on microtubule dynamics.

2. A second concern about the plus end growth rate analysis has to do with how correction of specimen drift was implemented. Appropriate drift correction is essential because specimen drift will otherwise be convolved with measurements of plus end growth. There are a couple of concerns:

First, the technique employed is not described well. As presented, there is not enough information to know exactly how timepoint to timepoint drift was calculated. It is stated that the average motion direction of all tracks at a given time point was calculated and smoothed over 10 frames. However, it is not stated how that calculation was performed nor exactly what smoothed means and how smoothing was implemented. I am guessing that the frame t to $t+1$ displacements for each detected spot were calculated, then the global mean averaged over frames t to $t+10$ to determine the drift vector for frame t to $t+1$. However, this is just a guess and the authors need to be more precise in their description in order for another experimenter to repeat the method.

Second, there is potential for a significant problem when correcting for specimen drift this way when imaging EB1 or other plus tip tracking labels in plant cortical arrays. When only the moving marker is employed for drift correction, drift correction methods assume that that particles movements are not correlated. In fact, there is often a net bias in the partitioning of microtubule orientation in ordered cortical arrays (described for example by Dixit et al MBOC 2006). When the only fiducials being used are the moving spots of plus end label, drift correction/registration algorithms can be dominated by this population of moving points, which have similar velocities and directions of movement. This can grossly distort the frame by frame transformation matrix used for drift correction and thus the measured particle velocities. In an extreme example, think of all the marked plus ends traveling in the same direction at the same velocity - drift correction would cause them to become fixed in place and to cause the cellular reference frame to move instead. It is necessary in these cases to use fiducials that are actually part of the reference frame to correct for drift. These can be spots of autofluorescence in cell walls or signal at the anticlinal walls and junctions of anticlinal walls. An additional concern here with drift correction is that there is a gradient of drift velocity along the same axis that a gradient in particle velocities is measured, with cells nearest the

root tip (the TZ) moving the most rapidly in the image frame and cells at the distal DZ moving the least or even being stationary. Further, the net orientation of plus end movement relative to the cell axis (and this the axis of greatest drift) and changes steadily from TZ to DZ. Thus, and errors or biases in drift correction could result in a gradient of effects on measured tip velocities from TZ to CZ in a complex way. It is hard to determine if there are any significant issues with drift correction from the data presented, but it is very important that the authors verify that drift correction has properly fixed the reference frame for particle movement in each root zone that was analyzed.

3. It is concluded that microtubule behaviours (orientation and plus end growth) were correlated with the gradient of cytokinin activity. Causality was then tested by both adding exogenous CK and by mutation of CK receptors. However, the receptor experiments seem incomplete because only the TZ and EZ were examined in these experiments. In mock treated wildtype roots, MT orientation is transverse in both the TZ and EZ, it is only the DZ which shows oblique and higher net orientations. Therefore, in the genetic experiments, loss of receptor function was only shown to attenuate response to addition of exogenous CK, these experiments did not provide evidence that loss of receptor function affects inherent microtubule orientation or MT growth tip growth rates. The conclusion stated at the end of this paragraph - "...cytokinin-mediated control of the CMTs network requires a functional AHK4/CRE1 receptor."- should be restated more precisely to refer to a requirement for the response to the addition of exogenous CK, not control of the CMT array in general. On the other hand, the DZ was examined in the context of array sensitivity to oryzalin, where receptor mutants were found to be more resistant to oryzalin. However, oryzalin resistance is not a measure of array orientation nor is it a direct measure of the status of inherent plus end growth rate.

4. The observed effects of cytokinin application and manipulation of cytokinin perception are interpreted as relatively directly effects of CK action - for example, that cytokinin "fine tunes" microtubules, or the statement in the paper's that cytokinin "guides microtubule activity". However, it is also possible that the observed effects of hormone addition on CMT array behavior are less direct, for example, lying downstream of changes in cell growth rather than being a direct target of hormone action. This was a conclusion reached in a paper the authors cite in their introduction on the effects of auxin on microtubule arrays in excised hypocotyl segments (Adamowski et al 2019), but was not discussed in the present manuscript as an alternative hypothesis. In the data presented in this manuscript, there appears to be a good correlation between cell growth state and CMT behavior. The authors observed that addition of cytokinin and auxin both slow down root growth, and that CK has large effects on both TZ and EZ cells while auxin has a large effect only on EZ cells. Interestingly, CK application also was observed to cause changes in MT behavior in TZ and EZ cells while auxin application primarily affected MT arrays in EZ cells. Thus, a further discussion and consideration of the relationships between hormone action, cell growth state and microtubule array behavior seems warranted.

5. There are a number of questions about the oryzalin experiments. First In setting up the experiments with oryzalin, the authors suggest that the observed decrease in microtubule plus end growth rates in the DZ indicated that these arrays are less dynamic and thus might be predicted to have reduced sensitivity to the action of oryzalin, thus they tested this prediction. A couple of comments here. First, only microtubule growth rates were measured by EB1-GFP labeling. The polymerization dynamics of these arrays is a function also of the shrinking rates and the transition frequencies between growth, pause and shrinking states. In addition, these arrays feature two ended dynamics as the most microtubules also have dynamics minus ends. While plus end growth was observed to be reduced upon cytokinin application, this does not necessarily mean that the microtubules are less dynamic on the whole. For example, ends could be growing more slowly but undergoing more rapid transitions between and shrinking. Second, it is not obvious a priori why

microtubules with more slowly growing ends would be less susceptible to the action of oryzalin, in fact, the opposite might be expected if it were only tubulin polymerizing in vitro. The observation that these polymers were more resistant to oryzalin may reflect the action of a stabilizing function inhibiting subunit loss (such as CLASP activity), rather than the slower polymerization rate.

Second, the data shown in figure 1F for oryzalin results are example images only, with no quantitation. Later in the manuscript, the reader is directed to Figure S4A, which does show quantitation of oryzalin experiments. This figure should be called out along with figure 1F. However, there are also questions with this supplemental figure. The quantitation here is of MAP4-GFP signal intensity, which is measured to drop after oryzalin treatment. The major difference to oryzalin application only is that cells in the DZ show decreased sensitivity. But in figure S4A, there is no measurement of MAP4-GFP signal prior to oryzalin application in the DZ cells in order to make a before and after comparison, there is data from oryzalin treated cells only. This value is in similar range as those for untreated TZ and DZ cells, but a proper experimental design requires measurement from cells in the DZ before treatment.

Third, intensity of MAP4-MBD is used as the metric for microtubule integrity in the experiments with oryzalin, but there is no mention about whether background subtraction was performed, and if so, how it was done. In principle, measurement of MAP4-GFP signal can be a good way to measure MT array integrity, but this approach can be complicated by the fact that MAP4-GFP signal is redistributed, rather than destroyed, by microtubule depolymerization. If the whole cell was imaged as a volume, signal intensity should therefore be about the same before and after depolymerization. The fact that MAP4-GFP signal intensity is observed to fall following depolymerization by oryzalin is likely due to redistribution of signal into parts of the cell that are not in the analyzed optical volume. How background subtraction is performed can have a big influence on what is measured.

6. As discussed in comment 5, there is a need for wider consideration of MT dynamics. Only one narrow aspect is measured here, plus end growth rates, yet conclusions are reached about array dynamics as a whole. Sweeping conclusions and statements about array dynamics from measurement of plus end growth rates only should be avoided.

7. Why isn't orientation in the DZ assessed in the cytokinin receptor mutants? If cytokinin fine tunes microtubule behaviors, as the authors contend, wouldn't reduced perception in the DZ, where pronounced changes in plus end growth and orientation were observed to occur in wildtype, be expected to result in altered MT orientation and dynamics?

8. The conclusions for the oryzalin experiments on receptor mutants were difficult to understand (lines 264-266). It is concluded that the results suggest that cytokinin receptors might contribute to configuration of CMT's but the results from the experiments with oryzalin did not address CMT orientation. It is also concluded that the receptors might contribute to adjustments of CMT pattern and dynamic s under fluctuating cytokinin levels. Again, the oryzalin experiments test arrays resistance to oryzalin, not pattern nor end dynamics per se.

9. (line 201) The authors state that kymograph analysis of Eb1-GFP was used to confirm the change in MT orientation observed after application of CK. Did the authors instead that brightest point projections of EB1-GFP were used as support? Kymographs are useful for assessing particle velocities, not orientation. It is further states that kymographs revealed that the change in EB1-GFP velocities after CK application and cites figure 2H along with figure 2G. Figure 2H shows velocities calculated from particle tracking with Trackmate, not from from kymographs. There is extremely minimal analysis of velocities from kymographs, with only one kymograph being shown for each

treatment. If the authors want to state that kymograph analysis robustly supports the particle tracking data, they need to show analysis of populations of EB1 tracks from images registered to account for cell drift.

10. In figure 2, two sets of images are shown of CK and IAA sensors to support the idea that the CK and IAA activity have complimentary gradients in epidermal cells along the root axis. However, it is not apparent in the images that these patterns are in fact complimentary. IAA signal is quite low in all nuclei and it is hard to discern. There is an attempt at quantitation in figure S3A, but no stats are provided. The gradient in IAA certainly does not look strong in this graph.

A variety of conclusions are made about sensor activity, cell identity (trichoblasts vs atrichoblasts) and trichoblast maturation. However, these conclusions are supported only by example images with no measurement nor statistical analysis.

Altogether, the sensor experiments data do not add much to the paper and could simply be deleted. If they are to remain, there needs to be adequate quantitation and statistical analysis.

11. It is an excellent idea to double check results with use of an antibody to assess array orientation. However, if the image quality shown in figure S1b is representative, it is a little hard to see how strong specific orientations were able to be measured, such as for IAA on the EZ. One can see from these images a clear difference between mock and treatment, but a clear new net orientation.

12. In the experiments where auxin and cytokinin are added together or in sequence, it is concluded that cytokinin might overrule iaa effects on CMT orientation in the EZ, as pretreatment with CK (over an extended time) makes the cells in the EZ insensitive to further orientational changes by the addition of auxin. Given that CK pretreatment for an extended time is needed, would this necessarily be "overruling" by CK activity, which implies some active function, or that the cells are now in a new state where they become insensitive to IAA activity for CMT orientational responses?

13. To support better the conclusion that cytokinin and auxin activities are strikingly different, quantitation of the sensor data should be presented, currently this conclusion is supported by a single merged image in S3B. This difference is the basis for the entire section from line 329 to 342. Further, the reported "random" arrangement of CMT in trichoblast cells is not measured and the single image shown (S9D) is not of good enough quality to assess MT orientation in the trichoblast well (at least by eye). MT's in this cell certainly appear differently than its neighbors, but are they random in orientation? Finally, it is concluded these results reveal a correlation between auxin/cytokinin ratio and CMT dynamics. It is hard to see how this is the case since dynamics were not measured nor was the CK ratio.

14. The observation that cytokinin application also reduces microtubule growth rates and decreases sensitivity to MT depolymerizing drugs in lymphocytes is very interesting. However, I would be cautious about concluding there is a conserved pathway until more molecular details are worked out. While it is true that tubulin is highly conserved, as are some MAPs, plants also have unique MAPs that are important for MT dynamics and signaling pathways may well be more divergent than core features of the cytoskeleton itself.

Minor comments:

Line 32 Not all cytoskeletal elements are tubular in structure.

Line 33 Cross out "all processes". That seems too broad for the range of functions for which the cytoskeleton is essential.

Line 45 "...the cytoskeleton goes through substantial rearrangements to accommodate cytophysiological changes that occur during cytogenesis.". This is a vague and rather full of jargon. Can the sentence be written in a more straightforward and precise way?

Line 50 The cell plate separates daughter cells, the phragmoplast helps to organize and assemble the cell plate.

Line 75 There are earlier citations for the effect of auxin on cytoskeletal organization in the plant axis (mostly in the shoot). These should be cited along with the author's previous paper (Chen et al.).

Line 135 oscillate implies a regular and repetitive fluctuation, "alternate" might be a better term

Line 301, what does it mean to test the mutual interplay?

The legend for S2 states S2 shows brightest point projection, but the images are totally dominated by the track overlays. The reader has to get to the bottom of the legend to learn that. I would either show the projections as a separate set of images or just show the track overlays with a new description.

Referee #3:

Montesinos and colleagues present fascinating insights into the role of the plant hormones cytokinin and auxin for the orientation of cortical microtubules in growing roots. This is a highly relevant topic, since MT orientation is the major determinant for anisotropic growth of plant cells, since they not only provide mechanical constraint, but also serve as a template for the deposition of cellulose fibrils in the cell wall. The authors show beautifully how MTs reorient during cell elongation and quantify the influence of cytokinin and auxin on three important subdomains of the root using treatments and genetic interference. The take home message is that cytokinin leads to a stabilization of MTs and promotes an oblique orientation and counteracts the influence of auxin, which had been shown before to induce active MT reorganization and longitudinal orientation. The authors carefully map the activity domains of auxin and cytokinin using state of the art reporters and show a clear correlation in MT behavior with cellular signaling state. Importantly, they also demonstrate that cytokinin preferentially acts via CRE1, one of the three known cytokinin receptors, while the role of the other appears only minor. Using this tool, the authors show evidence that cytokinin perception is required to counteract the effects of auxin on MTs during root differentiation. However, the mechanisms of this important interaction remain elusive.

Finally, the manuscript closes on data that demonstrate that cytokinin treatment can protect animal cells in tissue culture from the MT depolymerization effects of Nocodazole.

Overall, this is a very nice manuscript with beautifully presented and carefully quantified data. My only criticism relates to the lack of insight into the mechanisms of the cytokinin effect and the interaction with auxin. Since this interaction has been shown to occur a number of different levels,

from biosynthesis to negative feedback, I think it would be important to delineate whether the crosstalk happens up- or downstream of transcription of cytokinin target genes. Established tools to activate Type-B ARRs, such as ARR1-Delta-DDK or ARR1-Delta-DDK-GR should allow to draw these conclusions.

We thank the reviewers for their valuable comments and suggestions. We have expanded the existing results and incorporated new observations into the revised version of our manuscript, which we hope address and resolve all major concerns of reviewers.

Referee #1:

In this exciting manuscript Montesinos and colleagues study the effects of cytokinin on microtubule reorientation during root development. Utilizing the epidermal cells as model system, they first describe the orientation of MTs in the different developmental zone of the root. They show that in the TZ and EZ MTs are arranged transversally, whereas in the EZ obliquely. Subsequently they show that 1 h treatment of cytokinin is sufficient to promote MTs reorientation from transversal to obliquous. They also show that cytokinin dependent MTs regulation is largely reliant on the activity of the AHK4 cytokinin. They demonstrate that cytokinin counteracts the rapid auxin-induced MT re-arrangement in an AHK4 dependent manner at the TZ and DZ. Intriguingly they also show that atrichoblast cells have higher cytokinin activity and obliquous MTs whereas trichoblast cells show higher auxin activity and randomly arranged MTs, supporting their findings. Finally, utilizing leukocytes as model animal system, authors demonstrate that cytokinin activity on MT arrangement is conserved in animal cells.

Major points:

To this reviewer the study is a fundamental piece in cytokinin research; nonetheless there are some concerns to be addressed prior publication.

My major concern regards the requirement of AHK4 but not of AHK3 in the regulation of MT arrangement. Indeed, AHK4/CRE1 is a cytokinin receptor expressed in vasculature cells, whereas AHK3 is expressed also in epidermal cells. Authors show that cytokinin acts on epidermal cells in an AHK4 dependent manner. Is the activity of cytokinin on MT arrangement cell non autonomous? Please discuss.

Response: We thank the reviewer for the comment. The tissue-specific localization of the different cytokinin receptors is definitely a relevant aspect. Based on the promotor GUS reporter analyses expression of all three receptors including AHK4/CRE in the root vasculature has been reported (Nishimura *et al*, 2004; Bielach *et al*, 2012). In addition, tools like Arabidopsis eFP Browser (University of Toronto, <http://bar.utoronto.ca/efp/cgi-bin/efpWeb.cgi?dataSource=Root>), which provide gene expression data collected using fluorescence-activated cell sorting suggest that there is a considerable expression of both AHK3 and AHK4/CRE1 in epidermal cells at the root transition and elongation zone (*Revision Figure 1*).

As it concerns distinct contributions of AHK4 and AHK3 cytokinin receptor to fine-tuning of the CMTs, our experimental data point at the AHK4 as a major receptor involved in the regulation of CMTs cytoskeleton in epidermal cells at elevated cytokinins. Motivated by comments of the reviewers (see also response to the reviewer 2 point 7) we extended our analyses to observe CMTs in epidermal cells of cytokinin receptor mutants along the longitudinal root growth axis including in depth analysis of cells in the differentiation zone. Intriguingly, we found that in *ahk3-3* mutant there is a higher number of cells with transversal disposition of CMTs, suggesting that epidermal cells might enter differentiation phase later when compared to wild-type (Fig. EV4A). Hence, we hypothesize that AHK3 perception might contribute to fine-tuning of CMTs along the longitudinal root growth axis, while under conditions, which might lead to increase of endogenous levels of cytokinins (e.g. different type of stresses) the contribution of AHK4 receptor to fine-tuning of CMTs is most pronounced.

Revision Figure 1. Expression pattern of cytokinin receptors *CRE1/AHK4*, *AHK2* and *AHK3* in *Arabidopsis* roots according to the *Arabidopsis* eFP Browser data collection using fluorescence-activated cell sorting.

Moreover, in case of cytokinin cell non autonomous mechanism on MT arrangement, would it be possible that cytokinin causes rearrangement of MTs via auxin? If so, how could the dynamics of cytokinin and auxin activity on MT be explained?

Response: Cytokinin and auxin are two hormonal regulators that are intimately connected at many levels including mutual regulation of hormone biosynthesis, signaling and cytokinin impact on auxin transport. Thus, we cannot exclude that some aspects of the cytokinin effects on CMTs involve the cross-talk with auxin. On the other hand, our experiments reveal a number of distinct effects that are exhibited by these two hormones such as *i.*) auxin promotes rapid re-arrangements of CMTs into longitudinal orientation whereas cytokinin drives oblique disposition of CMTs; *ii.*) cytokinin, but not auxin, reduces CMTs growth at the plus-end and *iii.*) only cytokinin interferes with CMTs in leukocytes whereas no effect of auxin could be detected. These observations suggest that there might be a cytokinin specific effects. However, we agree that the CMTs might be an important convergence point in auxin- cytokinin regulation of plant development and growth. This is supported by our findings that cytokinin pre-treatment can prevent auxin driven longitudinal re-orientation of CMTs in wild-type, or that deficiency of cytokinin perception in *ahk4/cre1-12* affects

sensitivity of CMTs to auxin driven reorientation in longitudinal direction. We believe that dissecting cytokinin-auxin interaction in regulation of CMTs is important following step that would need further and deeper investigation, which we find is out of the main scope of our current manuscript.

Also, given that cytokinin promotes MT rearrangement in leukocytes and given that orthologues of two component systems are missing in animals, it would be plausible that cytokinin acts via mechanisms independent by the ARR. Analysis of MT arrangement after inductions of phospho-insensitive overexpressor of ARR1 (35S::ARR1DDK:GR) would be enough to demonstrate this point.

Response: We thank the reviewer for suggesting this experiment, which indeed would clarify part of the molecular mechanism underlying cytokinin mediated regulation of CMTs cytoskeleton. As suggested, we used the inducible overexpressor line 35S::ARR1DDK-GR, in which after application of dexamethasone the constitutively active version of the transcription factor ARR1 is translocated to the nucleus where it regulates gene expression (Sakai *et al*, 2000, 2001) (Fig. 4A). We found that ARR1DDK promoted re-orientation of CMTs to oblique orientation in epidermal cells at the TZ and EZ already 3 hours after induction by DEX (Fig. 4B,C). Furthermore, activation of ARR1DDK-GR in root tips reduced sensitivity to oryzalin triggered swelling when compared to wild-type, and this effect was further accentuated when the ARR1DDK induction was accompanied with external cytokinin treatment (Fig. 4D-F). Thus, the constitutively active ARR1 mimicked effects of cytokinin on CMTs. In addition, we complemented the study about the role of ARR1 in the cytokinin mediated regulation of CMTs by analyzing the *arr1-3* mutant allele. The absence of the ARR1 transcription factor affected sensitivity of CMTs cytoskeleton to cytokinin (Fig. EV3D,E). Similarly to *ahk4/cre1-12*, in *arr1-3* mutant cytokinin was not that effective in preventing the root swelling triggered by oryzalin and most of root tips of *arr1-3* exhibited severe root swelling also in presence of cytokinin (Fig. EV3A,C).

Altogether, these results suggest that ARR1 is involved in the transmission of the cytokinin signal to regulate the CMTs dynamics in the root and suggest that convergence of pathways coordinating CMTs cytoskeleton in plant and in animal cell presumably occurs more downstream.

It is not clear to me the contribution given by the cytokinin dependent orientation of MT to root development. To test the activity in root development authors reports that plants treated with cytokinin and oryzalin for 72 hours show dampened root growth inhibition effects than plant treated with only oryzalin, however I do not consider this data sufficient to demonstrate the role of cytokinin dependent MT reorientation in development.

Response: We appreciate this comment. We wish to clarify that both treatments either oryzalin, or oryzalin in combination with cytokinin dampen root growth comparably (Fig. EV3C). However, while oryzalin application results in complete disintegration of CMTs cytoskeleton and massive swelling of cells at the root tip, the cytokinin pre- and co-treatment can significantly attenuate both these oryzalin effects. Based on these observation we conclude that cytokinin interference with oryzalin effects on the root tip might not be related to its impact on root growth. However, we agree that this is rather indirect indication that cytokinin through effects on CMTs might interfere with oryzalin treatment. Considering raised criticism we revised the manuscript accordingly.

It is known that cytokinin treatments inhibit root growth after 12 hours of treatment and that in plants exposed to transient cytokinin treatments (transferring plants after sufficient time for root inhibition from media supplemented with cytokinin to media without cytokinin), root restart growing after few hours of transfer. Knowing whether the re-establishment of growth after cytokinin

fluctuations is preceded by a re-orientation of MT from obliquous to transversal at the TZ and DZ would represent permit to better understand the developmental importance of the cytokinin mediated MT rearrangement.

Response: We thank the reviewer for bringing to our attention this very interesting point. To address it we monitored CMTs in cells at the EZ of roots, which were mock or cytokinin pretreated and afterwards transferred to the mock medium. We found that re-orientation of CMTs in cells at the TZ triggered by cytokinin is not reversible and CMTs remain in oblique orientation during five hours of observation. Neither recovery of root growth was observed after removal of cytokinins hinting at non-transient nature of changes of CMTs driven by cytokinins that might correlate with premature transition of cells to differentiation stage (Fig. EV1I, J; Movie EV7,8). We speculate that presumably longer time (more than 6 hours) after removal of cytokinin is required to generate new pool of epidermal cells whose elongation lead to recovery of root growth kinetics that has been reported previously (Márquez *et al*, 2019).

Minor comments:

I believe that the manuscript would benefit from a graphical model reporting the activity of auxin and cytokinin on MT arrangement in the different zones of the root.

Response: We appreciate this suggestion and following the advice we prepared a scheme of the cytokinin effect on the CMTs that we used for the synopsis of the paper.

It is not clear from the text how the TCS::GFP gradient was demonstrated. To this reviewer it looks like the considered gradiend only refer to the one in the TZ, DZ and EZ of the epidermis. Please specify.

Response: We fully agree that the original presentation of the cytokinin and auxin response gradients in roots were not sufficiently clear. In the revised manuscript we performed more thorough monitoring and analyses of auxin and cytokinin reporters. At least 18 roots were used for monitoring of auxin and cytokinin reporters in epidermal cells at the meristematic zone (very last 4 cells), TZ, EZ and DZ. Intensity of reporter signals was quantified and statistically evaluated (Fig. 2A-B).

Referee #2:

How signaling pathways that control plant growth interface with the core machinery of growth, such as the cytoskeletal, secretory and cell wall biosynthetic machinery, is one of the major frontiers in our understanding of plant growth and function. In organisms with rigid cell walls, including higher plants, a cortical cytoskeleton guides cell wall biosynthesis and cell wall expansion, shaping the growth and morphogenesis of the organism. Here, the authors investigate the relationship of cytokinin signaling with the behavior and organization of the microtubule cytoskeleton in roots of *Arabidopsis thaliana*. An increasing gradient of cytokinin signaling activity has been described from the root meristem to the zone where cells derived from the meristem finally mature and differentiate. The authors describe a correlation of cortical microtubule organization and growth rates of microtubule plus end with this gradient of cytokinin activity (the organizational gradient has been described in previous studies). They then test causation by exogenous application of cytokinin and by genetic disruption of cytokinin receptors. In addition, they investigate possible interactions of the auxin and cytokinin biosynthetic pathways in modulating cytoskeletal behavior and they

demonstrate that exogenous application of cytokinin can also perturb plus end growth in animal tissue culture cells. Major conclusions are that cytokinin tunes cytoskeletal organization and behavior across along the root axis, that auxin and cytokinin may affect cytoskeletal behavior and cell growth by different but interacting pathways, and that plant and animal cells may share a conserved mechanism of action in modulating microtubule growth.

The manuscript is lengthy but well organized, describing a large number of experiments. In general, a strength of the study is a quantitative approach to the questions being addressed. It is excellent that a vertically oriented confocal instrument was used to image root and cellular behavior, avoiding confounding hormonal signaling with gravitropic signaling. This was impressive and is seldom done. The manipulation of cytokinin activity by use of mutants in cytokinin receptors was also a strength, being an important complement to the use of exogenous hormone application, which may not always probe normal physiology well. On the other hand, most major effects of cytokinin application on the cytoskeleton were observed with or depended on exogenous hormone application, including the consequences of receptor mutation. Defects in cytoskeletal organization and plus end growth rates were not observed when receptor function only was perturbed.

Response: We agree with the reviewer comments, although we would like to highlight correlation observed between level of cytokinin response monitored by TCS reporter and CMT dynamics; e.g. we observed that in cells in the DZ, which exhibit higher cytokinin response, CMTs are in oblique orientation and display reduced plus-end growth when compared to cells in TZ and EZ. Similarly, CMTs arrangement is distinct in trichoblasts when compared to atrichoblasts both exhibiting different levels of cytokinin signaling output (please note quantification of CMTs orientation in revised manuscript Fig. EV4C).

We agree, that we did not detect any dramatic alterations of CMTs cytoskeleton in cytokinin receptor mutants, unless challenged by cytokinins. Inspired by the reviewer suggestion we paid more attention to the role of cytokinin receptor functions in regulation of CMTs in cells located more proximally at the DZ. Intriguingly, we found that in *ahk3-3* mutant number of cells with transversal orientation of CMTs along the longitudinal root growth axis is increased, when compared to wild-type. Furthermore, we observed that in epidermal cells at the beginning of the DZ (for these purposes we defined the start of the DZ as a zone where bulging of root hairs in trichoblasts is detected, please see also response to the point 7) CMTs orientation was perfectly oblique in wild-type while still random in *ahk3-3* mutant. These observations hint at contribution of the AHK3 receptor to fine-tuning of CMTs along the root growth axis (Fig EV4B in the revised manuscript).

In addition I had a number of other comments and concerns about the experiments and their interpretation, as presented below.

1. A major part of the study is the measurement of microtubule growth rates as measured by EB1-GFP labeling. A systematic analysis of plus end microtubule growth in the plant root growth zone has not previously been reported. One issue with expression of EB1 which is not mentioned, but should be discussed in the text, is that EB1 overexpression changes microtubule dynamics - speeding up growth rates and stabilizing the growth phase. In the context of this study, it would be good to know if the observed pattern of plus end growth rates along the root axis is sensitive to EB1 expression itself. The possible influence of EB1 expression on patterns of plus end growth rates might be assessed by performing kymograph analysis on cells expressing fluorescent protein (FP) fusions to tubulin (using suitably registered datasets). Use of FP-tubulin fusions is preferable to using lines

expressing MAP4-GFP as expression of MAP4-GFP also has significant effects on microtubule dynamics.

Response: We thank the reviewer for relevant comment and useful suggestions. We agree that GFP reporter might affect both localization and dynamics of the EB1b-GFP fusion protein as well as the activity of CMTs. As reported by (Skube *et al*, 2010) in animal cells, GFP tag might affect ability of EB1 to bind MTs, or to modify interaction of other MAPs with the MTs. However, in this study they also provide evidences that unlike the N-terminally tagged GFP-EB1 reporter, fusion of GFP to the C-terminal end of EB1 interferes less with the functionality of EB1 and binding to MTs. We would like to stress that for our study we used C-terminal fusion of EB1b-GFP construct driven by EB1b native promoter, which might limit anomalies that are possibly caused by high accumulation of N-terminal GFP-EB1 in overexpressor lines.

Furthermore, following the suggestion of the reviewer, we also monitored CMTs using the *35S::mCherry-TUA5* reporter and performed additional Kymograph analysis. For that, we used the Dragonfly spinning disk microscope, recently acquired at our institute. We also took advantage of the open access tool Kymobutler (Jakobs *et al*, 2019), which allows evaluation of several CMTs plus-end trajectories. Observation of the CMTs visualized with mCherry-TUA5 during 5 minutes after 1 hour pre-treatment with mock or cytokinin supplemented medium supported our previous observation (Fig. EV1H). Analyses using Kymobutler showed that the CMTs growth rate in epidermal cells in the EZ is $2.30 \pm 0.03 \mu\text{m}/\text{min}$ in mock conditions, and $1.80 \pm 0.03 \mu\text{m}/\text{min}$ after 1 hour cytokinin treatment. These values are very similar to the CMTs plus-end growth rates obtained by analyzing the EB1b-GFP protein trajectories ($2.40 \pm 0.07 \mu\text{m}/\text{min}$ in mock, and $1.74 \pm 0.06 \mu\text{m}/\text{min}$ after 1 hour cytokinin treatment).

2. A second concern about the plus end growth rate analysis has to do with how correction of specimen drift was implemented. Appropriate drift correction is essential because specimen drift will otherwise be convolved with measurements of plus end growth. There are a couple of concerns:

First, the technique employed is not described well. As presented, there is not enough information to know exactly how timepoint to timepoint drift was calculated. It is stated that the average motion direction of all tracks at a given time point was calculated and smoothed over 10 frames. However, it is not stated how that calculation was performed nor exactly what smoothed means and how smoothing was implemented. I am guessing that the frame t to $t+1$ displacements for each detected spot were calculated, then the global mean averaged over frames t to $t+10$ to determine the drift vector for frame t to $t+1$. However, this is just a guess and the authors need to be more precise in their description in order for another experimenter to repeat the method.

Response: We value this comment and we paid attention to provide more details about the applied technique. Briefly, at each time point the frame to frame displacements for all detected spots were averaged. The resulting drift vector of the sample is then smoothed using a moving average filter with a span of 10 time frames. We apologies for the insufficient description of the analysis procedure, and we incorporate a more detailed protocol in the corresponding section of the material and methods.

Second, there is potential for a significant problem when correcting for specimen drift this way when imaging EB1 or other plus tip tracking labels in plant cortical arrays. When only the moving marker is employed for drift correction, drift correction methods assume that that particles movements are not correlated. In fact, there is often a net bias in the partitioning of microtubule orientation in

ordered cortical arrays (described for example by Dixit et al MBOC 2006). When the only fiducials being used are the moving spots of plus end label, drift correction/registration algorithms can be dominated by this population of moving points, which have similar velocities and directions of movement. This can grossly distort the frame by frame transformation matrix used for drift correction and thus the measured particle velocities. In an extreme example, think of all the marked plus ends traveling in the same direction at the same velocity - drift correction would cause them to become fixed in place and to cause the cellular reference frame to move instead. It is necessary in these cases to use fiducials that are actually part of the reference frame to correct for drift. These can be spots of autofluorescence in cell walls or signal at the antinodal walls and junctions of antinodal walls. An additional concern here with drift correction is that there is a gradient of drift velocity along the same axis that a gradient in particle velocities is measured, with cells nearest the root tip (the TZ) moving the most rapidly in the image frame and cells at the distal DZ moving the least or even being stationary. Further, the net orientation of plus end movement relative to the cell axis (and this the axis of greatest drift) and changes steadily from TZ to DZ. Thus, any errors or biases in drift correction could result in a gradient of effects on measured tip velocities from TZ to CZ in a complex way. It is hard to determine if there are any significant issues with drift correction from the data presented, but it is very important that the authors verify that drift correction has properly fixed the reference frame for particle movement in each root zone that was analyzed.

Response: The concerns related to drift and possible impact on the evaluation of CMTs kinetics raised by reviewers are very relevant. This was one of the important issues to be solved when setting up imaging for the CMTs plus-end growth rate in cells, which are moving. For that, the following steps were taken to address this potential problem:

1. EB1b-GFP trajectories were evaluated in videos of individual epidermal cells (either TZ, EZ or DZ). This approach reduces substantially collateral miscalculations due to the movement of the cell or to the drift correction, correcting this drift movement specifically for the root zone, which is evaluated in every case. Therefore, the results are not influenced by the movement of the rest of the root. (Furthermore, the rest of the root is not taken into account for the stabilization).
2. In cases the FIJI Plugins "Stack Reg" or "Correct 3D drift" were able to stabilize the movie correctly the output of the plus-end growth rate analysis with and without prior stabilization was compared and found in good agreement. The calculated drift of the pre-stabilized movies was ~ 0 as expected.
3. The plus-end growth rate analysis script outputs the drift corrected movies for visual inspection.
4. In all movies presented the calculate sample drift is 2 orders of magnitude lower than plus end growth rate. Hence, disabling the drift correction has a negligible effect on the results.

Attached, the reviewer can find an example of all the point mentioned before, in a EB1b-GFP video of one epidermal cell in the EZ, including: (i) original movie (no stabilization) (revision movie 1) (ii) movie after Stack Reg stabilization (revision movie 2), (iii) the plus end drift and the plus-end drift applied to the movie (revision movie 3), and (iv) the comparison of the CMTs plus-end growth speed with and without the Drift Stabilization (Revision Figure 2).

Finally, we would also like to point out, that as is visible in all EB1b-GFP movies, trajectories of EB1b-GFP particles are not following a unique direction, but several different trajectories. Therefore, drift correction/registration algorithms cannot be dominated by this population of moving points.

In summary, the evaluation of the EB1b-GFP trajectories was performed considering all the potential problems derived from the drift and the different cell speed movement in the different root zones.

Detailed description of the analysis of the EB1b-GFP is now included in the corresponding section of the material and methods.

Revision Figure 2. Optimization of the EB1b-GFP speed trajectories evaluation. (i) Time lapse movie (5 minutes) of epidermal cell of the EZ expressing *EB1b::EB1b-GFP* (Revision Movie 1). (ii) Stabilization of Revision Movie 1 with Stack reg stabilization (Revision Movie 2). (iii) Drift correction over time estimated for Revision Movie 1 (left panel) and the movie obtained after applying the Drift correction estimated (Revision Movie 3, right panel). (iv) Quantification of the speed EB1b-GFP trajectories (µm/min) without (upper table) and with the Drift Stabilization (lower table).

3. It is concluded that microtubule behaviours (orientation and plus end growth) were correlated with the gradient of cytokinin activity. Causality was then tested by both adding exogenous CK and by mutation of CK receptors. However, the receptor experiments seem incomplete because only the TZ and EZ were examined in these experiments. In mock treated wildtype roots, MT orientation is transverse in both the TZ and EZ, it is only the DZ which shows oblique and higher net orientations. Therefore, in the genetic experiments, loss of receptor function was only shown to attenuate response to addition of exogenous CK, these experiments did not provide evidence that loss of receptor function affects inherent microtubule orientation or MT growth tip growth rates. The conclusion stated at the end of this paragraph - "...cytokinin-mediated control of the CMTs network requires a functional AHK4/CRE1 receptor."- should be restated more precisely to refer to a requirement for the response to the addition of exogenous CK, not control of the CMT array in general. On the other hand, the DZ was examined in the context of array sensitivity to oryzalin, where receptor mutants were found to be more resistant to oryzalin. However, oryzalin resistance is not a measure of array orientation nor is it a direct measure of the status of inherent plus end growth rate.

Response: These are all relevant points.

Firstly, we would like to stress that correlation between cytokinin activity and CMTs is based on monitoring of the cytokinin reporter in roots grown on standard Murashige and Skoog (MS) medium, without exogenous cytokinins. The *TCSn::ntdT-tNOS* expression profile exhibited gradual increase across the TZ and the EZ toward the DZ, which correlates with rearrangements of CMTs from transversal in cells of the TZ and EZ to oblique in cells of the DZ (Fig. 2A, B).

However, we agree with the reviewer that there are no dramatic changes in CMTs orientation and plus end growth observed in cytokinin receptor mutants unless they are exposed to exogenous cytokinins. We hypothesize that in normal growth conditions lack of one receptor is compensated by other members of the family. However, lack of one receptor activity might become more critical under conditions that lead to increase of endogenous cytokinin levels, e.g. in plants exposed to biotic or abiotic stresses (Hare *et al*, 1997; Argueso *et al*, 2009; O'Brien & Benková, 2013). Our data suggests that in such a conditions the AHK4/CRE1 receptor might play a dominant function in fine-tuning the CMTs. As far as it concerns other receptors, our new data (please see response to the point 7) hint at AHK3 involvement in fine-tuning of CMTs as part of molecular network controlling CMTs in cells along the longitudinal growth axis (Fig. EV4A). We admit that conclusions about the role of cytokinin receptors should be better formulated to avoid over-interpretation, and we revised respective paragraphs of the manuscript.

4. The observed effects of cytokinin application and manipulation of cytokinin perception are interpreted as relatively directly effects of CK action - for example, that cytokinin "fine tunes" microtubules, or the statement in the paper's that cytokinin "guides microtubule activity". However, it is also possible that the observed effects of hormone addition on CMT array behavior are less direct, for example, lying downstream of changes in cell growth rather than being a direct target of hormone action. This was a conclusion reached in a paper the authors cite in their introduction on the effects of auxin on microtubule arrays in excised hypocotyl segments (Adamowski *et al* 2019), but was not discussed in the present manuscript as an alternative hypothesis. In the data presented in this manuscript, there appears to be a good correlation between cell growth state and CMT behavior. The authors observed that addition of cytokinin and auxin both slow down root growth, and that CK has large effects on both TZ and EZ cells while auxin has a large effect only on EZ cells. Interestingly, CK application also was observed to cause changes in MT behavior in TZ and EZ cells while auxin application primarily affected MT arrays in EZ cells. Thus, a further discussion and consideration of the relationships between hormone action, cell growth state and microtubule array behavior seems warranted.

Response:

We agree with the reviewer, that effect of cytokinin on CMTs dynamics and cell growth cannot be disregarded and we cannot exclude that part of the observed effects are also related to cell growth control. On other hand, there are observations, which hint at more direct effects of cytokinins, unrelated to cell growth regulation: i.) both auxin and cytokinin inhibits cell expansion, but auxin does not exhibit any dramatic effects on plus end growth (unlike cytokinin), ii.) cytokinin, but not auxin, affects MTs in animal cells, in which effects mediated through cell expansion are unlikely.

5. There are a number of questions about the oryzalin experiments. First In setting up the experiments with oryzalin, the authors suggest that the observed decrease in microtubule plus end growth rates in the DZ indicated that these arrays are less dynamic and thus might be predicted to have reduced sensitivity to the action of oryzalin, thus they tested this prediction. A couple of

comments here. First, only microtubule growth rates were measured by EB1-GFP labeling. The polymerization dynamics of these arrays is a function also of the shrinking rates and the transition frequencies between growth, pause and shrinking states. In addition, these arrays feature two ended dynamics as the most microtubules also have dynamics minus ends. While plus end growth was observed to be reduced upon cytokinin application, this does not necessarily mean that the microtubules are less dynamic on the whole. For example, ends could be growing more slowly but undergoing more rapid transitions between and shrinking. Second, it is not obvious a priori why microtubules with more slowly growing ends would be less susceptible to the action of oryzalin, in fact, the opposite might be expected if it were only tubulin polymerizing *in vitro*. The observation that these polymers were more resistant to oryzalin may reflect the action of a stabilizing function inhibiting subunit loss (such as CLASP activity), rather than the slower polymerization rate.

Response: We agree that application of oryzalin to test cytokinin effect on MTs are based on some assumptions, such as that by cytokinin altered dynamics of CMTs might interfere with oryzalin triggered depolymerisation of CMTs. For example it has been demonstrated that taxol stabilized microtubules are only partially sensitive to oryzalin (Morejohn *et al*, 1987; Hugdahl & Morejohn, 1993). However, molecular pathway that downstream of cytokinin perception fine-tunes CMTs activity remains to be uncovered. It is plausible that activity of some microtubule associated proteins (such as MAP65s or CLASPs) is affected by cytokinin, which leads to observed changes in CMTs cytoskeleton. Our ongoing work is focused on identification and functional characterization of the factors involved in regulation of CMTs by cytokinin.

Second, the data shown in figure 1F for oryzalin results are example images only, with no quantitation. Later in the manuscript, the reader is directed to Figure S4A, which does show quantitation of oryzalin experiments. This figure should be called out along with figure 1F.

Response: We agree with the reviewer suggestion. In the revised manuscript, we re-arranged the figure panel to present both images and quantification of CMTs in epidermal cells of all root growth zones after oryzalin and combined cytokinin/oryzalin treatments in wild-type (Fig. 1G) and cytokinin receptor mutants (Fig. 3J, Appendix Fig. S3D).

However, there are also questions with this supplemental figure. The quantitation here is of MAP4-GFP signal intensity, which is measured to drop after oryzalin treatment. The major difference to oryzalin application only is that cells in the DZ show decreased sensitivity. But in figure S4A, there is no measurement of MAP4-GFP signal prior to oryzalin application in the DZ cells in order to make a before and after comparison, there is data from oryzalin treated cells only. This value is in similar range as those for untreated TZ and DZ cells, but a proper experimental design requires measurement from cells in the DZ before treatment.

Response: We noticed there is a need to clarify some technical details related to the quantification of MAP4-GFP in cells treated with oryzalin. For every treatment (either mock, oryzalin or cytokinin/oryzalin), and for every root zone (TZ, EZ or DZ), the CMTs intensity was measured at time 0 and at time 60 min. Values presented in graphs (Fig. 1G, 3J, Appendix Fig. S3D) are ratios of fluorescence detected at t_{60}/t_0 . As mentioned above, in the revised manuscript we analyzed sensitivity of CMTs to oryzalin at the DZ of all cytokinin receptor mutants (Fig. 1G, 3J, Appendix Fig. S3D).

Third, intensity of MAP4-MBD is used as the metric for microtubule integrity in the experiments with oryzalin, but there is no mention about whether background subtraction was performed, and if so, how it was done. In principle, measurement of MAP4-GFP signal can be a good way to measure MT array integrity, but this approach can be complicated by the fact that MAP4-GFP signal is redistributed, rather than destroyed, by microtubule depolymerization. If the whole cell was imaged as a volume, signal intensity should therefore be about the same before and after depolymerization. The fact that MAP4-GFP signal intensity is observed to fall following depolymerization by oryzalin is likely due to redistribution of signal into parts of the cell that are not in the analyzed optical volume. How background subtraction is performed can have a big influence on what is measured.

Response: We agree with the reviewer that quantification of the CMTs array integrity requires specific and careful way of evaluation. We would like to clarify that we did not use any background subtraction procedure in the files we present in the manuscript. To limit problems derived from the redistribution of the MAP4-GFP signal (no-polymerizing dimers), we performed a maximum projections using z-stacks that includes only the cortical area (proximal to the plasma membrane). In this way, we detected primarily signal associated with the CMTs, and thereby we limited signal of non-bound MAP4-GFP in cytosol. No other adjustments were performed on the images presented in the manuscript. We included the detailed imaging protocol in methods.

6. As discussed in comment 5, there is a need for wider consideration of MT dynamics. Only one narrow aspect is measured here, plus end growth rates, yet conclusions are reached about array dynamics as a whole. Sweeping conclusions and statements about array dynamics from measurement of plus end growth rates only should be avoided.

Response: We appreciate the reviewer's comment. We focused on disposition of CMTs and the plus-end growth as a standard measure for evaluating CMTs cytoskeleton. We fully agree that monitoring of additional parameters would be needed to describe dynamics of CMTs cytoskeleton. Considering the raised criticism we modified all respective statements to avoid misinterpretation.

7. Why isn't orientation in the DZ assessed in the cytokinin receptor mutants? If cytokinin fine tunes microtubule behaviors, as the authors contend, wouldn't reduced perception in the DZ, where pronounced changes in plus end growth and orientation were observed to occur in wildtype, be expected to result in altered MT orientation and dynamics?

Response: We agree that closer analysis of the DZ might provide more insights into role of cytokinin receptors in regulation of CMTs. In this context we find critical to define criteria that unequivocally define DZ. Typically, in wild-type roots, the DZ can be defined as the area where the epidermal cells stop to elongate (Verbelen *et al*, 2006; Pavelescu *et al*, 2018); CMTs re-orient to oblique disposition and growth rate of CMTs at the plus-end is reduced (Fig. 1B,E). Another hallmark of cell transition to differentiation stage might be formation of root hairs in the trichoblast cell file (Verbelen *et al*, 2006). Based on the observed correlation between gradient of cytokinin response and CMTs arrangements along the longitudinal root growth axis we hypothesized that lack of cytokinin perception might also affect progression of cells from TZ and EZ with transversal orientation of CMTs to differentiation status characterized by oblique disposition of the CMTs. Therefore we decided to examine whether lack of perception mediated through cytokinin receptor mutants interfere with transition of cells to differentiation stage by scoring number of epidermal cells with transversal orientation of CMTs along the longitudinal root growth axis. We found that while in *cre1-12/ahk4*

mutant the number of root epidermal cells with transversal disposition of the CMTs was not different from wild-type control (10.13±1.11 epidermal cells for WT and 10.12±0.95 epidermal cells for *cre1-12/ahk4*; Fig. EV4A), in the *ahk3-3* mutant significantly higher number of cells with transversal CMTs orientation (11.92±1.09 epidermal cells) when compared to wild-type was detected. Furthermore, when as an indicator of the DZ bulging of root hairs was considered, as expected the CMTs in the atrichoblast cell file in wild-type roots exhibited oblique disposition (41.70 ± 15.0 degrees) and similarly, no significant changes in CMTs orientation in *cre1-12/ahk4* when compared to wild-type were detected (41.09 ± 13.53 degrees) (Fig. EV4B,C). Interestingly, in the *ahk3-3* the CMTs with average orientation of CMTs 51.96 ± 18.75 degrees was significantly different from wild type suggested more transversal orientation (Fig. EV4B,C).

Based on these results we hypothesize that AHK3 perception might contribute to fine-tuning of CMTs along the longitudinal root growth axis, while under conditions which might lead to increase of endogenous levels of cytokinins (e.g. different type of stresses) the contribution of AHK4 receptor to fine-tuning of CMTs appears to be more prominent.

8. The conclusions for the oryzalin experiments on receptor mutants were difficult to understand (lines 264-266). It is concluded that the results suggest that cytokinin receptors might contribute to configuration of CMT's but the results from the experiments with oryzalin did not address CMT orientation. It is also concluded that the receptors might contribute to adjustments of CMT pattern and dynamics under fluctuating cytokinin levels. Again, the oryzalin experiments test arrays resistance to oryzalin, not pattern nor end dynamics per se.

Response: We apologize that analyses of cytokinin receptor mutant sensitivity to cytokinin were not clearly presented and discussed. The main motivation to perform these experiments was based on finding that in wild-type roots cytokinin pre-treatment can attenuate oryzalin triggered depolymerization of CMTs. As already discussed above, (response to the point 5) this we understand as supportive evidence that cytokinin might affect CMTs. To explore and gain more experimental support for cytokinin effects on CMTs we tested a role of cytokinin receptors. These data suggested that cytokinin to attenuate oryzalin mediated depolymerisation of CMTs requires functional AHK4/CRE1 receptor. We agree that these assay do not provide information about cytokinin effects on CMTs dynamics and pattern. To avoid any misleading interpretations we carefully revised respective parts of the manuscript.

9. (line 201) The authors state that kymograph analysis of Eb1-GFP was used to confirm the change in MT orientation observed after application of CK. Did the authors instead that brightest point projections of EB1-GFP were used as support? Kymographs are useful for assessing particle velocities, not orientation.

Response: We agree with the reviewer that Kymograph cannot be used to address particle orientation, we apologize for the impreciseness. By monitoring of the MTs at plus end using EB1b-GFP reporter for 30 seconds by maximum projection we noticed that in different cell types we detect not only different velocities of EB1-GFP signal suggesting distinct kinetics of plus-end growth, but also different directions of trajectories detected by tracking of EB1b-GFP signal. Taking into consideration the comment of the reviewer we replaced the term kymograph for manual particle tracking, which we find to be more accurate description.

For the manual particle tracking, the selection of the EB1b-GFP positive dots was random (not including only the brightest signals). The trajectory of the individual EB1b-GFP signals was monitored

over the time during 60 seconds. Estimating the distance walked by the particle we calculated the speed, which in all cases fitted with the CMTs plus-end growth rate quantified (Fig. 1D, 2I and 3G). We decided to modify the term Kymograph of the Figures 1D, 2I and 3F to EB1b-GFP manual particle tracking, as a supporting visual material of the EB1b-GFP particle movement analysis.

Additionally, as suggested by the reviewer, in the revised manuscript we included an additional quantification of the CMTs growth, using *35S::mCherry-TUA5* line and Kymograph analysis (Fig. EV1H). The analysis using mCherry-TUA5 reporter fully supported observations based on the EB1b-GFP particle tracking analysis.

It is further states that kymographs revealed that the change in EB1-GFP velocities after CK application and cites figure 2H along with figure 2G. Figure 2H shows velocities calculated from particle tracking with Trackmate, not from from kymographs. There is extremely minimal analysis of velocities from kymographs, with only one kymograph being shown for each treatment. If the authors want to state that kymograph analysis robustly supports the particle tracking data, they need to show analysis of populations of EB1 tracks from images registered to account for cell drift.

Response: Quantification of CMTs plus-end growth rate (presented at Figures 1D, 2I and 3G) was based on analyses of EB1b-GFP signal trajectories over the time as described in the Material and Method (using Trackmate and custom script). Kymograph (or now renamed to EB1b-GFP manual particle tracking) were not used for quantifications but only for purposes of better visual presentation of CMTs plus-end growth. We agree that results based on EB1b-GFP manual particle tracking (“kymographs) were not sufficiently robust and therefore we modified text accordingly.

10. In figure 2, two sets of images are shown of CK and IAA sensors to support the idea that the CK and IAA activity have complimentary gradients in epidermal cells along the root axis. However, it is not apparent in the images that these patterns are in fact complimentary. IAA signal is quite low in all nuclei and it is hard to discern. There is an attempt at quantitation in figure S3A, but no stats are provided. The gradient in IAA certainly does not look strong in this graph.

Response: We thank the reviewer for this comment. We agree that using term complementary, for profiles of auxin and cytokinin response might be misleading and we modified the text to better describe pattern of both pathways along the root axis. In the revised manuscript we performed thorough quantification of both auxin (*DR5v2::3nGFP*) and cytokinin (*TCSn::ntdT-tNOS*) reporters along the longitudinal root growth axis. Expression profiles of cytokinin and auxin reporters were monitored in epidermis including last 4 cells of the meristematic zone, cells at the TZ, EZ and DZ in 18 roots at minimum (Fig. 2B). We found that while increase of auxin response along the root axis followed very shallow gradient, expression *TCSn::ntdT-tNOS* exhibited gradual increase across the TZ and the EZ toward the DZ (Fig. 2B), in correlation with different arrangements of CMTs in these root zones (Fig 1B).

Technical comments: Nuclear GFP signal for the Auxin signaling reporter *DR5v2::3nGFP* is clearly higher in trichoblast epidermal cells than in atrichoblast epidermal cells (Fig. EV2A,B). We agree with the reviewer that it is difficult to discern the GFP signal in root tip epidermal cells, but it is presumably due to a lower signaling in this area, rather than as a consequence of an inefficient functionality of the reporter (as mentioned, in trichoblast signal of *DR5v2::3nGFP* reporter is strong).

A variety of conclusions are made about sensor activity, cell identity (trichoblasts vs atrichoblasts)

and trichoblast maturation. However, these conclusions are supported only by example images with no measurement nor statistical analysis.

Response: In the revised manuscript we included a quantification of the relative cytokinin and auxin response levels in atrichoblast and trichoblast cells in the DZ. To this end we calculated a ratio between *TCSn::ntdT-tNOS* and *DR5v2::3nGFP* signals. We observed that ratio is significantly higher in atrichoblast cells (prevailing cytokinin response) than in trichoblast epidermal cells, where the ratio is below 1, indicating higher auxin response when compared to that of cytokinin (Fig. EV2D).

Altogether, the sensor experiments data do not add much to the paper and could simply be deleted. If they are to remain, there needs to be adequate quantitation and statistical analysis.

Response: We believe that the visualization of cytokinin or auxin gradients along the root axis is important in context of the presented data. We agree that quantification and statistical evaluation of reporters signal was not sufficiently elaborated in the previous version of the manuscript. As mentioned above (point 10) in the revised manuscript, expression profiles of cytokinin and auxin reporters were monitored in epidermis including last 4 cells of the meristematic zone, cells at the TZ, EZ and DZ in 18 roots at minimum and results were statistically evaluated (Fig. 2B).

11. It is an excellent idea to double check results with use of an antibody to assess array orientation. However, if the image quality shown in figure S1b is representative, it is a little hard to see how strong specific orientations were able to be measured, such as for IAA on the EZ. One can see from these images a clear difference between mock and treatment, but a clear new net orientation.

Response: As immunodetection of CMTs in cells at TZ and EZ is technically very challenging it affects also quality of imaging in these cells. We repeated immunolocalisation and applied protocol that limits damaging of cells in more proximal root zones (Figure S1b (now Fig. EV1B)). In order to have a better and clearer picture, we would also like to share with the reviewer the way of how the CMTs orientation was measured in these images (we include several examples, the region of interest selected in cell, and the outcome (blue line) which is translated in angle degrees (Revision Figure 3)).

Revision Figure 3. CMTs orientation measurements using FibrilTool plugin (ImageJ). CMTs orientation quantification of root epidermal cells (immunostained using anti-tubulin antibodies) of TZ treated with mock (left panel) or CK (BAP 10 μ M 1h, right panel); or epidermal cell of EZ treated with Auxin (NAA 0.1 μ M 1h, middle panel). The region of interest (ROI) is delimited by the area with yellow border. The blue line indicates the average CMTs orientation in the selected ROI. Scale bar 10 μ m.

12. In the experiments where auxin and cytokinin are added together or in sequence, it is concluded that cytokinin might overrule *iaa* effects on CMT orientation in the EZ, as pretreatment with CK (over an extended time) makes the cells in the EZ insensitive to further orientational changes by the addition of auxin. Given that CK pretreatment for an extended time is needed, would this necessarily be "overruling" by CK activity, which implies some active function, or that the cells are now in a new state where they become insensitive to IAA activity for CMT orientational responses?

Response: We agree with the reviewer that "overruling" might not be the proper way to describe the effect of cytokinin preventing the Auxin effect on the CMTs. We thank for the note and we modified the text in the revised version. What we suggest is that once the cells perceive increased cytokinin signal the cell differentiation program would be triggered that include the adjustment of the CMTs dynamics. Once the cell reaches this status, the sensitivity of CMTs to hormonal regulation by auxin might be attenuated. In other scenarios, for example applying simultaneously auxin and cytokinin,

auxin is able to induce the re-orientation of the CMTs to longitudinal (Appendix Fig. S5A), probably because cytokinin requires at least 30 min to perform its effect (Appendix Fig. S5B,C).

13. To support better the conclusion that cytokinin and auxin activities are strikingly different, quantitation of the sensor data should be presented, currently this conclusion is supported by a single merged image in S3B. This difference is the basis for the entire section from line 329 to 342. Further, the reported "random" arrangement of CMT in trichoblast cells is not measured and the single image shown (S9D) is not of good enough quality to assess MT orientation in the trichoblast well (at least by eye). MT's in this cell certainly appear differently than its neighbors, but are they random in orientation? Finally, it is concluded these results reveal a correlation between auxin/cytokinin ratio and CMT dynamics. It is hard to see how this is the case since dynamics were not measured nor was the CK ratio.

Response: We thank the reviewer for the comments and suggestions with respect to the epidermal trichoblast cells. In the revised version we extended the study for in depth analyses of trichoblasts including thorough monitoring of CMTs and expression of auxin and cytokinin reporters.

Using MAP4-GFP and EB1-GFP reporter we show that orientation of the CMTs in trichoblast cells is predominantly longitudinal (21.17 ± 23.3 degrees), and significantly different when compared to the orientation of CMTs in atrichoblast cells in the same area (Fig. EV4C, Movie EV9). As described in our response to the point 10, we also quantified cytokinin and auxin signaling levels in trichoblast and atrichoblast cells. Reduced *TCSn::ntdT-tNOS / DR5v2::3nGFP* ratio in trichoblast versus atrichoblast cells points at higher auxin activity in trichoblasts when compare to atrichoblast cells (Fig. EV2D). We believe that this dataset better support our conclusion related to correlation between auxin/cytokinin ratio and CMTs.

14. The observation that cytokinin application also reduces microtubule growth rates and decreases sensitivity to MT depolymerizing drugs in lymphocytes is very interesting. However, I would be cautious about concluding there is a conserved pathway until more molecular details are worked out. While it is true that tubulin is highly conserved, as are some MAPs, plants also have unique MAPs that are important for MT dynamics and signaling pathways may well be more divergent than core features of the cytoskeleton itself.

Response: We appreciate the comment. Cytokinin is reducing the MTs plus-end growth rate and increasing the resistance of MTs to de-polymerizing drugs in lymphocytes cells. We consider these results a potential starting point to study possible conserved mechanism among animal and plants in terms of MTs dynamics regulation. Indeed, currently we focus our investigation on studying other molecular players involved in the MTs dynamics regulation, and more specifically on MAPs conserved in both kingdoms.

Minor comments:

Line 32 Not all cytoskeletal elements are tubular in structure.

We corrected it.

Line 33 Cross out "all processes". That seems too broad for the range of functions for which the cytoskeleton is essential.

We modified it.

Line 45 "...the cytoskeleton goes through substantial rearrangements to accommodate cyto-physiological changes that occur during cytogenesis.". This is a vague and rather full of jargon. Can the sentence be written in a more straightforward and precise way?

Line 50 The cell plate separates daughter cells, the phragmoplast helps to organize and assemble the cell plate.

We fixed it.

Line 75 There are earlier citations for the effect of auxin on cytoskeletal organization in the plant axis (mostly in the shoot). These should be cited along with the author's previous paper (Chen et al.).

We also included other references (Nick *et al*, 1992; Takesue & Shibaoka, 1998; Takahashi *et al*, 2003; Le *et al*, 2005; Vineyard *et al*, 2013; True & Shaw, 2020).

Line 135 oscillate implies a regular and repetitive fluctuation, "alternate" might be a better term

We changed it.

Line 301, what does it mean to test the mutual interplay?

We meant to examine interaction between auxin and cytokinin in regulation of CMTs. The sentence was modified accordingly.

The legend for S2 states S2 shows brightest point projection, but the images are totally dominated by the track overlays. The reader has to get to the bottom of the legend to learn that. I would either show the projections as a separate set of images or just show the track overlays with a new description.

We modified the legend of the figure (now Fig. EV1C-G).

Referee #3:

Montesinos and colleagues present fascinating insights into the role of the plant hormones cytokinin and auxin for the orientation of cortical microtubules in growing roots. This is a highly relevant topic, since MT orientation is the major determinant for anisotropic growth of plant cells, since they not only provide mechanical constraint, but also serve as a template for the deposition of cellulose fibrils in the cell wall. The authors show beautifully how MTs reorient during cell elongation and quantify the influence of cytokinin and auxin on three important subdomains of the root using treatments and genetic interference. The take home message is that cytokinin leads to a stabilization of MTs and promotes an oblique orientation and counteracts the influence of auxin, which had been shown before to induce active MT reorganization and longitudinal orientation. The authors carefully map the activity domains of auxin and cytokinin using state of the art reporters and show a clear correlation in MT behavior with cellular signaling state. Importantly, they also demonstrate that cytokinin preferentially acts via CRE1, one of the three known cytokinin receptors, while the role of the other appears only minor. Using this tool, the authors show evidence that cytokinin perception is required to counteract the effects of auxin on MTs during root differentiation. However, the

mechanisms of this important interaction remain elusive.

Finally, the manuscript closes on data that demonstrate that cytokinin treatment can protect animal cells in tissue culture from the MT depolymerization effects of Nocodazole.

Overall, this is a very nice manuscript with beautifully presented and carefully quantified data. My only criticism relates to the lack of insight into the mechanisms of the cytokinin effect and the interaction with auxin. Since this interaction has been shown to occur a number of different levels, from biosynthesis to negative feedback, I think it would be important to delineate whether the crosstalk happens up- or downstream of transcription of cytokinin target genes. Established tools to activate Type-B ARR, such as ARR1-Delta-DDK or ARR1-Delta-DDK-GR should allow to draw these conclusions.

Response: We thank the reviewer for the comments and suggestion (also brought by the reviewer1), which indeed would clarify part of the molecular mechanism underlying cytokinin mediated regulation of CMTs cytoskeleton. As suggested, we used the inducible overexpressor line *35S::ARR1ΔDDK-GR*, in which after application of dexamethasone the constitutively active version of the transcription factor ARR1 is translocated to the nucleus where it regulates gene expression (Sakai *et al*, 2000, 2001) (Fig. 4A). We found that ARR1ΔDDK promoted re-orientation of CMTs to oblique orientation in epidermal cells at the TZ and EZ already 3 hours after induction by DEX (Fig. 4B,C). Furthermore, activation of *ARR1ΔDDK-GR* in root tips reduced sensitivity to oryzalin triggered swelling when compared to wild-type, and this effect was further accentuated when the *ARR1ΔDDK* induction was accompanied with external cytokinin treatment (Fig. 4D-F). Thus, the constitutively active ARR1 mimicked effects of cytokinin on CMTs. In addition, we complemented the study about the role of ARR1 in the cytokinin mediated regulation of CMTs by analyzing the *arr1-3* mutant allele. The absence of the ARR1 transcription factor affected sensitivity of CMTs cytoskeleton to cytokinin (Fig. EV3D,E). Similarly to *ahk4/cre1-12*, in *arr1-3* mutant cytokinin was not that effective in preventing the root swelling triggered by oryzalin and most of root tips of *arr1-3* exhibited severe root swelling also in presence of cytokinin (Fig. EV3A,C). Furthermore, we also observed that in the *arr1-3* mutant, pre-treatment with cytokinin cannot prevent the auxin driven re-orientation of CMTs to longitudinal disposition as observed for wild-type (Fig EV1B compared to Fig. EV3D,E).

Altogether, these results suggest that ARR1 is involved in the transmission of the cytokinin signal to regulate the CMTs dynamics in the root and suggest that convergence of pathways coordinating CMTs cytoskeleton in plant and in animal cell presumably occurs more downstream.

REFERENCES:

- Argueso CT, Ferreira FJ & Kieber JJ (2009) Environmental perception avenues: The interaction of cytokinin and environmental response pathways. *Plant, Cell Environ.* **32**: 1147–1160
- Bielach A, Podlešáková K, Marhavý P, Duclercq J, Cuesta C, Müller B, Grunewald W, Tarkowski P & Benková E (2012) Spatiotemporal regulation of lateral root organogenesis in Arabidopsis by cytokinin. *Plant Cell* **24**: 3967–3981
- Hare PD, Cress WA & Van Staden J (1997) The involvement of cytokinins in plant responses to environmental stress. *Plant Growth Regul.* **23**: 79–103
- Hugdahl JD & Morejohn LC (1993) Rapid and reversible high-affinity binding of the dinitroaniline herbicide oryzalin to tubulin from *Zea mays* L. *Plant Physiol.* **102**: 725–740
- Jakobs MA, Dimitracopoulos A & Franze K (2019) Kymobutler, a deep learning software for automated kymograph analysis. *Elife* **8**: 1–19

- Le J, Vandenbussche F, De Cnodder T, Van Der Straeten D & Verbelen JP (2005) Cell elongation and microtubule behavior in the Arabidopsis hypocotyl: Responses to ethylene and auxin. *J. Plant Growth Regul.* **24**: 166–178
- Márquez G, Alarcón MV & Salguero J (2019) Cytokinin Inhibits Lateral Root Development at the Earliest Stages of Lateral Root Primordium Initiation in Maize Primary Root. *J. Plant Growth Regul.* **38**: 83–92 Available at: <http://dx.doi.org/10.1007/s00344-018-9811-1>
- Morejohn LC, Bureau TE, Mole-Bajer J, Bajer AS & Fosket DE (1987) Oryzalin, a dinitroaniline herbicide, binds to plant tubulin and inhibits microtubule polymerization in vitro. *Planta* **172**: 252–264
- Nick P, Schäfer E & Furuya M (1992) Auxin redistribution during first positive phototropism in corn coleoptiles: Microtubule Reorientation and the Cholodny-Went theory. *Plant Physiol.* **99**: 1302–1308
- Nishimura C, Ohashi Y, Sato S, Kato T, Tabata S & Ueguchi C (2004) Histidine kinase homologs that act as cytokinin receptors possess overlapping functions in the regulation of shoot and root growth in arabidopsis. *Plant Cell* **16**: 1365–1377
- O'Brien JA & Benková E (2013) Cytokinin cross-talking during biotic and abiotic stress responses. *Front. Plant Sci.* **4**: 1–11
- Pavelescu I, Vilarrasa-Blasi J, Planas-Riverola A, González-García M, Caño-Delgado AI & Ibañes M (2018) A Sizer model for cell differentiation in Arabidopsis thaliana root growth. *Mol. Syst. Biol.* **14**: 1–18
- Sakai H, Aoyama T & Oka A (2000) Arabidopsis ARR1 and ARR2 response regulators operate as transcriptional activators. *Plant J.* **24**: 703–711
- Sakai H, Honma T, Takashi A, Sato S, Kato T, Tabata S & Oka A (2001) ARR1, a transcription factor for genes immediately responsive to cytokinins. *Science (80-.).* **294**: 1519–1521
- Skube SB, Chaverri JM & Goodson H V. (2010) Effect of GFP tags on the localization of EB1 and EB1 fragments in vivo. *Cytoskeleton* **67**: 1–12
- Takahashi H, Kawahara A & Inoue Y (2003) Ethylene Promotes the Induction by Auxin of the Cortical Microtubule Randomization Required for Low-pH-Induced Root Hair Initiation in Lettuce (*Lactuca sativa* L.) Seedlings. *Plant Cell Physiol.* **44**: 932–940
- Takesue K & Shibaoka H (1998) The cyclic reorientation of cortical microtubules in epidermal cells of azuki bean epicotyls: The role of actin filaments in the progression of the cycle. *Planta* **205**: 539–546
- True JH & Shaw SL (2020) Exogenous Auxin Induces Transverse Microtubule Arrays Through TRANSPORT INHIBITOR RESPONSE1/AUXIN SIGNALING F-BOX Receptors. *Plant Physiol.* **182**: 892–907
- Verbelen JP, De Cnodder T, Le J, Vissenberg K & Baluška F (2006) The root apex of arabidopsis thaliana consists of four distinct zones of growth activities: Meristematic zone, transition zone, fast elongation zone and growth terminating zone. *Plant Signal. Behav.* **1**: 296–304
- Vineyard L, Elliott A, Dhingra S, Lucas JR & Shaw SL (2013) Progressive transverse microtubule array organization in hormone-induced Arabidopsis hypocotyl cells. *Plant Cell* **25**: 662–676

Thank you for submitting a revised version of your manuscript. Your study has now been seen by two of the original referees, who find that their main concerns have been addressed and are now in favour of publication of the manuscript. There now remain only a couple of minor editorial issues that have to be addressed before I can extend formal acceptance of the manuscript.

Please let me know if you have any further questions regarding any of these points. You can use the link below to upload the revised files.

Thank you again for giving us the chance to consider your manuscript for The EMBO Journal. I am looking forward to receiving the final version.

Referee #1:

Montesinos and colleagues replied to most of my comments.

minor point:

In figure 4 E statistical treatments are missing.

Referee #3:

I would like to congratulate the authors on a truly impressive revision! My point on the lack of insight into how cytokinin affects MT orientation has been addressed more than adequately. The experiments contrasting activated ARR1 with the loss of function situation are clear cut and the addition of oryzalin co-treatments simply stunning.

I strongly support publication.

The authors performed the requested changes.

Editor accepted the revised manuscript.

Corresponding Author Name: Eva Benkova

Journal Submitted to: The EMBO Journal

Manuscript Number: EMBOJ-2019-104238